# Role of Antioxidant Natural Products in Management of Infertility: A Review of Their Medicinal Potential

**DOI:** 10.3390/antiox9100957

**Published:** 2020-10-07

**Authors:** Seungjin Noh, Ara Go, Da Bin Kim, Minjeong Park, Hee Won Jeon, Bonglee Kim

**Affiliations:** 1College of Korean Medicine, Kyung Hee University, Hoegi-dong Dongdaemun-gu, Seoul 05253, Korea; nohapril@khu.ac.kr (S.N.); a_ra0328@khu.ac.kr (A.G.); cr7switness@khu.ac.kr (D.B.K.); pmj6543224@khu.ac.kr (M.P.); 2Department of Pathology, College of Korean Medicine, Kyung Hee University, Hoegi-dong Dongdaemun-gu, Seoul 05253, Korea; titmal@khu.ac.kr; 3Korean Medicine-Based Drug Repositioning Cancer Research Center, College of Korean Medicine, Kyung Hee University, Hoegi-dong Dongdaemun-gu, Seoul 05253, Korea

**Keywords:** infertility, natural products, plant extract, anti-inflammation, antioxidants

## Abstract

Infertility, a couple’s inability to conceive after one year of unprotected regular intercourse, is an important issue in the world. The use of natural products in the treatment of infertility has been considered as a possible alternative to conventional therapies. The present study aimed to investigate the effects and the mechanisms of various natural products on infertility. We collected articles regarding infertility and natural products using the research databases PubMed and Google Scholar. Several natural products possess antioxidant properties and androgenic activities on productive factors and hormones. Antioxidants are the first defense barrier against free radicals produced by oxidative stress (OS). They remove reactive oxygen stress (ROS), reducing insulin resistance, total cholesterol, fat accumulation, and cancer growth. Moreover, various natural products increase endometrial receptivity and fertility ability showing androgenic activities on productive factors and hormones. For example, *Angelica keiskei* powder and *Astragalus mongholicus* extract showed anti-infertility efficacies in males and females, respectively. On the other hand, adverse effects and acute toxicity of natural products were also reported. *Tripterygium glycoside* decreased fertility ability both in males and females. Results indicate that management of infertility with natural products could be beneficial with further clinical trials to evaluate the safety and effect.

## 1. Infertility

Accumulating evidence indicates that the prevalence of human infertility has increased over the past decades [1]. Oxidative stress (OS) is one of the major causes of defective generative function [2]. OS, which arises from an imbalance between reactive oxygen species (ROS) and protective antioxidants, influences the entire reproductive lifespan of men and women [3]. At controlled levels, OS facilitates some physiological reproductive functions but at higher levels it is implicated in pathological processes in the reproductive tract that contribute to infertility and poor pregnancy outcomes. Some studies have addressed the possible relationship between reproductive toxicity and exposure to ROS or its chemical sources [4]. Elevated levels of ROS directly damage oocytes and sperm DNA and induce apoptosis in sperm [3,5]. In male reproductive system, ROS disrupts the integrity of the sperm DNA and contributes to lipid peroxidation [4,6]. Biological membranes, like sperm’s, are particularly vulnerable to ROS effects [7]. Excessive production of ROS in reproductive tract can both damage the fluidity of sperm plasma membrane and the integrity of DNA in sperm nucleus, which can lead to spermatogenesis dysfunction and toxic effects on sperms, causing the lipid peroxidation of sperm membrane [8]. Such peroxidation exercises deterious influences and leads to serious pathological changes and infertility [9]. Differentiation of the sperms is dependent on the Leydig cells which secrete testosterone hormone [10]. Degeneration of the Leydig cells induced a reduction in testosterone production and spermatogenesis arrest. A previous longitudinal study on age related changes in serum testosterone levels showed that the incidence of hypogonadism in men increased with age [11]. The production of testosterone is decreased in aged Leydig cells, because of cellular changes in the steroidogenic pathway that decrease the production of testosterone, decrease luteinizing hormone (LH) stimulated cyclic adenosine monophosphate (cAMP) production, and downregulate steroidogenic acute regulatory protein (StAR), CYP11A1 in the mitochondria, and CYP17A1 in the smooth endoplasmic reticulum (ER).

Female infertility can be caused by failures at various steps, including ovulation, fertilization, embryo development, embryo transport, and implantation [12]. The different responses of environment toxicity include reduced fertility, spontaneous abortions, low birth weight, impaired folliculogenesis, and even damage to the ovaries [13]. OS induces infertility in woman through a variety of mechanisms [3], having a direct effect on the oocyte, embryo, and implantation by causing cell membrane lipid peroxidation, cellular protein oxidation, and DNA damage [14]. Excess ROS in the follicle may overwhelm follicular fluid antioxidant defense and hinder the endometrium which normally functions to support the embryo and its development [3]. Appropriate development of embryo and receptive endometrium are crucial factors for successful implantation [15]. Endometrial receptivity is critical for blastocyst adhesion and invasion during the complex process of implantation. Leukemia inhibitory factor (LIF) in particular is one of the major factors that regulates endometrial receptivity. Defects of LIF expression is involved in multiple implantation failures in patients with female infertility. OS is also associated with conditions such as endometriosis, hydrosalpinges, polycystic ovary syndrome (PCOS), and unexplained subfertility [14]. In addition, there is a lack of specific genetic markers because of the absence of an inherited syndrome that could implicate a gene in the pathogenesis of female infertility [16]. Mutations in the human LH P-subunit gene recently have been reported and linked with infertility. Endometriosis is noted in up to 30–40% of infertile women. Luteinizing hormone (LH) and its receptors have been linked with endometriosis-associated infertility. Reproduction is tightly controlled by hypothalamic–pituitary–gonadal axis [17]. Reproduction systems respond to hormonal signals from the pituitary gland which, in turn, is controlled by hormones produced in the hypothalamus [18]. Interruption of these processes, in any of the functional events in either sex, leads to fertility impairment [17] including gonadal dysgenesis, amenorrhea, premature ovarian failure [18]. Mammalian reproductive physiology is primarily regulated by the gonadotrophins luteinizing hormone (LH) and follicle stimulating hormone (FSH) secreted from the anterior pituitary which act on the gonads to produce sex steroids [19]. These pituitary hormones in turn enhance the proliferation of the follicular cells and the production of estrogens (principally estradiol) by ovarian cholesterol catabolism [20]. Additionally, they can lead to and restore spermatogenesis [21]. The initiation and maintenance of mammalian infertility are connected with G-protein coupled receptor 54 (GPR54) [22]. The mutation in GPR 54 is characterized by the absence of sexual maturation and low levels of gonadotropin releasing hormones (GnRH). Abnormal GnRH secretion induces anovulation, luteal insufficiency, and premature oocyte maturation, leading to menstrual disorders, polycystic ovary syndrome (PCOS), recurrent miscarriage, and infertility [16]. Additionally, it could affect the testicular function with decrease in T release [23]. Estrogen affects granulosa cells by promotion of proliferation, suppression of apoptosis, and augmentation of FSH effects. Homeostatic maintenance of prolactine (PRL) is essential since this hormone performs multiple physiological functions [24]. Increased PRL levels can cause infertility and bone loss in both women and men. It has been reported that E2 increases serum and pituitary PRL in ovariectomized rats. In addition, precursor of E2 and P4, pregnelone sulfate also increases prolactin production in the rat pituitary. On the other hand, compounds derived from natural food and herbal medicine showing promising antioxidant and antiapoptotic potentials have been considered an alternative therapy for disease [25].

The antioxidant system plays an importance role in protecting reproductive and other biological tissues below a critical threshold of ROS, preventing negative effects on reproduction [26]. Herbal medicines possessing antioxidants reduced ROS levels, protecting germ cells from OS-mediated apoptosis [27]. They could be used as complementary, alternative medicines to promote pregnancy [28].

### Infertility and Oxidative Stress

ROS contributes to inflammation and it is one of the causes of infertility [22]. Long-term high oxidative stress and chronic inflammation affect the reproductive system. Lipid peroxidation products in testis were determined by measuring malondialdehyde (MDA) [29]. Increased MDA level suggests severe oxidative damages and could cause infertility. An increased level of TNF-α is an indicator of inflammation [22]. *Nuclear factor kappa-light-chain-enhancer of activated B cells* (NF-*κB*) is known as the important mediator of inflammation. The expression of NF-*κB* leads to cell dysfunction and cell death. The activation of NF-*κB* by oxidative stress stimulates the proinflammatory response, upregulation of endothelin, and apoptosis. Proinflammatory cytokines such as IL-6 and TNF-α upregulate the expression of the suppress or of cytokine signaling 3 (SOCS3) implicated in inflammation-mediated insulin resistance in the liver and adipocytes. Testis is a highly prolific tissue with fast cellular renewal system along with rich in polyunsaturated fatty acids and ROS scavenging enzymes of low concentration in sperms’ plasmalemma, and for these reasons it becomes an easy target for the radiation-induced free radicals mediated damage [30]. Pathogenic bacteria cause inflammation and histopathological lesions in endometrium and perturb uterine involution, ovulation, and embryonic survival [31]. Bacterial endometritis is associated with toll like receptor 4 (TLR-4) complex signaling process and secretion of chemokines and cytokines including tumor necrosis factor (TNF-α), interleukin 1A and 6 (IL 1A and IL6). TNF-α and IL1A stimulate gene expression for potent chemotactic factors (IL-8), monocyte chemotactic protein-1 (MCP-1), and adhesion molecules on vascular endothelial cells leading to PMN recruitment to the site of inflammation and mobilization of neutrophils followed by phagocytosis of invading pathogens within the uterine lumen. Endometrial cells express TLR-4 for recognition of the lipopolysaccharide endotoxin of Gram-negative bacteria, leading to secretion of IL-6, IL-8, and prostaglandin E2.

## 2. Plant Extract and Infertility

Several herbal extracts and plant-derived pure molecules have shown their protective effects in various types of diseases [32], including those that affect the reproductive system [33]. Recent studies have shown that the administration of plant extracts improve semen parameters, androgen status, fertility index, and have positive influence on sperm quality in male [5,34]. In female, herbal medicine affects the molecular mechanism and prevents estrogen-dependent endometrial hyperplasia improving ovarian dysfunction, ovarian follicle [35], and increased endometrial receptivity [12,15]. Additionally, herbal therapy that has actions on the hypothalamic–pituitary–gonadal axis may influence reproductive physiology and ameliorates some infertility problems [17]. The gonadotrophic-like effects of the extracts were characterized by the following biological parameters: increase in the weight of the ovary and uterus; induction of ovulation; increase in estradiol, progesterone, protein levels; decrease in cholesterol level, and so forth [20]. The antimutagenic or protective effects have been attributed to many classes of phytocompounds mainly flavonoids and phenolic compounds [36]. The natural antioxidants with free radical scavenging ability have received much attention as potential remedies to treat oxidative stress and abnormal hormone functions [18,37]. Antioxidants can directly scavenge ROS, inactivate them, and repair the damage [14]. Additionally, they showed diverse biological activities resulting from their ability to mimic endogenous estrogen actions, inhibit hormone actions, and modulate hormone productions [18]. The antioxidant capacity of phenolic compounds, flavonoids, and foods rich in these compounds, has been repeatedly demonstrated in various in vitro and in vivo systems. In this present study, we aimed to investigate the effects and mechanisms of various plants extracts and natural products on the reproductive system. A large number of plants have been used to treat infertility for thousands years worldwide, including Korea [38,39]. Additionally, numerous natural products, including plant extracts were discovered to possess potential effects in reversing reproductive activity in both males and females. Natural products originated from plants, animals, and fungi, and their forms varied from compounds, extracts, as well as multiple formulas. Studies have discovered structural and functional improvements in the reproductive system while identifying the specific mechanisms of effects. However, adverse effects were also observed to be related with the utilization of some natural products.

### 2.1. Natural Products That Reverse Male Infertility

#### 2.1.1. Plant Derived Natural Products for Treatment of Male Infertility

##### In Vitro Studies

Several studies reported the efficacy of plant derived natural products through in vitro examination (Table 1). Date palm pollen extract from *Phoenix dactylifera* Linn. containing retina, cholesterol, and estrogenic compounds can stimulate gonadotropin activities [40]. Mahaldashtian et al. demonstrated that the administration of date palm pollen extract at doses of 0.06, 0.25, 0.62 mg/mL for 14 days to Sertoli and spermatogonial stem cells from mice elevated the number of spermatogonial colonies. The results identified that date palm pollen extract could be clinically used to enable the proliferation and differentiation of sperm cells. Jung et al. reported that 5H-purin-6-amine at doses of 0.01, 0.1, 1, and 10 µg/mL originated from *Sedum sarmentosum*, was an effective compound inducing spermatogonial stem cells proliferation [41]. Spermatogonial stem cells (SSC) from C57BL/6-TG-EGFP were cultured for 1 week with 1 µg/mL 5H-purin-6-amine which was identified as n-butanol fraction (Bu) of *Sedum sarmentosum*. This administration of this compound increased the Ki67, PLZF, GFRα1, VASA, Lhx1 levels, and decreased the level Pgk2. It was demonstrated that enhanced SSC proliferation may be attributed to the treatment of *Sedum sarmentosum*, treating male infertility. *Glycyrrhiza uralensis* Fisch. which is also called as licorice, is a well-known herb possessing cell proliferation and detoxification effects [42]. Wang et al. identified that licorice extract at doses of 0.2, 2, 20 μM for 72 h proliferated spermatogonia in testis tissue from C57BL/6N mice. Expression of BrdU staining positive cells and PCNA protein increased, along with improved levels of SCP3 and Spo11, which are proteins related to spermatocyte differentiation. Yang et al. elucidated that *Lycium barbarum* polysaccharide (50 μg/mL for 48 h) mediated the endoplasmic reticulum stress (ERS) pathway in Leydig MLTC-1 cells which were toxified by cisplatin (DDP) [43]. The results showed elevation in cell viability, testosterone, and ERS mediated protein expression (p-PERK/PERK, p-elF2α/elF2α, and ATF4/β-actin level), as well as inhibition of apoptotic factors (caspase 3, 7, and 12) and autophagy parameters (MDC-positive cell, LC3II/I, and Atg5/β-actin). Chang et al. reported that *Morindae officinalis* possessed a protective effect with a decreased lipid peroxidation level and increased testosterone production [44]. TM3 cells and mouse Leydig cells were incubated with *Morinda officinalis* aqueous extract at doses of 5, 10, 50, 100, 250 μg/mL and 100 μmol H_2_O_2_ for 24 h. The administration of the compound protected the cells from H_2_O_2_ induced cytotoxicity and lipid peroxidation, leading to increased testosterone production. Moreover, it showed increased expression of SOD, CAT, and mRNA, while there was a decrease in the level of MDA. These findings indicated that *Morindae officinalis* had protective effects on testosterone production. Chung et al. reported that *Taraxacum officinale* known as dandelion increased the levels of steroidogenic enzymes involved in the production of testosterone in the testis [11]. TM3 mouse Leydig cells, derived from ATCC No CRL mouse testes were treated with 1, 10, 25, 50 µg/mL of *Taraxacum officinale* aqueous extract for 48 h with fetal-bovine serum in Dulbecco’s modified eagle’s medium. It showed that the extract significantly activated the steroidogenic genes STAR, CYP11A1, and CYP17A1 in the smooth endoplasmic reticulum and increased their mRNA levels, thereby increasing the testosterone levels in mouse Leydig cells. These results indicated that the *Taraxacum officinale* may be used as alternative medicine for the treatment of diseases characterized by insufficient testosterone, such as male infertility. *Typha capensis* (Rohrb.) N.E.Br. belongs to the family of Typhaceae and is widely used for treating genital problems, male potency, and blood circulation [45]. Its root extract, treated with doses of 10 and 100 μg/mL for 96 h, improved testosterone production and marked significant difference in DNA fragmentation in TM3-leydig cells. This demonstrated that *Typha capensis* has potential effects on reversing infertility through mediation of testosterone.

##### In Vivo Studies

Numerous in vivo studies have reported the efficacy of plant derived natural products to enhance fertilization (Table 2). *Acacia hydaspica* R. Parker belongs to the family of Leguminosae, possessing antioxidant, anticancer, antihemolytic, anti-inflammatory, and analgesic properties [46]. *Acacia hydaspica* ethyl acetate extract (400 mg/kg for 21 days) was administrated to cisplatin induced SD rats. The results showed increases of seminiferous tubule diameter, area, and epithelial height, along with decreases of width of tubular lumen, interstitial space, and DNA damage (percentage of head DNA and tail DNA, tail movement, and comet length). Moreover, sex hormones (testosterone, LH, FSH), testicular tissue antioxidant enzymes (SOD, POD, CAT, QR, GSH, GR, GST, GPx, γ-GT) were significantly elevated, while the reactive species of H_2_O_2_, NO, MDA were downregulated. These all together described the protective potential of the natural compound against oxidative stress and testicular injuries. Salahipour et al. described the protective effects of *Achillea millefolium* inflorescences alcoholic extract at a dose of 120 mg/kg for 48 days against nicotine induced reproductive failure in Wistar mice [47]. The extract of *Achillea millefolium* Linn., which is known for its antioxidant and anti-inflammatory properties, enhanced sperm motility, capsule thickness, epithelial thickness, and tubule differentiation. Histological results marked improvements in epithelial and interstitial tissue morphology as well as the level of spermatogenesis. The restoration of levels of LH, LDH, SOD, MDA, and NO demonstrated the capacity of this natural product to recover toxic impacts of nicotine to male reproductive systems. Nasr et al. elucidated the testicular cytotoxic effects of aged garlic extract, originated from *Allium sativum* for Pekinense, which is known for its beneficial organo-sulfar compounds [10]. The aged garlic extract recovered the adriamycin induced testicular changes including low testis weight, low sperm count, low motility, thick irregular basal lamina of seminiferous tubules, and sperm abnormality.

Endocrine parameters and antioxidant enzymes (CAT, SOD, and MDA) were also regulated. Histopathological results showed that testis regained its normal structure, seminiferous tubules were organized, and Sertoli cells, primary spermatocytes, spermatids, and spermatozoa recovered their normal morphology. Bae et al. elucidated the profertility effects of decursin, extracted from the roots of *Angelica gigas* Nakai, which is well-known for its role to inhibit the growth of cancer cells [1]. TM3 Leydig cells treated with decursin revealed an upregulation of cell viability against H_2_O_2_ at doses of 10, 50 μg/mL for 2 h. SD rats were rendered infertile with cryptorchidism, then treated with the extract at a dose of 400 mg/kg for 4 weeks. It was found that left testicular weight, sperm count, percentage of motile spermatozoa, and spermatogenic cell density increased. Biochemical results suggested the possibility of decursin as an infertility treating supplement by exhibiting regulation in the level of antioxidant proteins (SOD, Nrf2, HO-1, and d 8-OHdG), and apoptotic factors (Bcl-2 and Bax). Basha et al. investigated the protective role of ginseng and banana leaf extract at a dose of 150 mg/kg for 30 days against testicular damage in streptozotocin mediated diabetic Swiss mice, which was aggravated by induction of fluoride [37]. These two extracts were each isolated from *Panax ginseng C. A. mey* and *Lagerstroemia speciosa* (L.) Pers., which are famous traditional Chinese medicines used for hyperglycemia, while ginseng is also known for its neuroprotective, anti-inflammatory, antiatherosclerotic effects. Each compound regulated sperm parameters (weight of testis, epididymis, and seminal vesicle, sperm density, viability, progressive motility, and morphology), content of glycogen, fructose, protein, and cholesterol in reproductive organs. Especially, coadministration of the two products exhibited better effectiveness in reproductive function. Ze-QingWu et al. reported that bajijiasu, which was isolated from the roots of *Morinda officinalis* F. C. could improve male reproductive capacity [6]. *Morinda officinalis* F.C., a famous herb in south China, has been used traditionally to treat kidney-yang-deficient diseases in China, including male infertility. The experiment was conducted in a normal group (20, 40, 80, 160, 320 mg/kg orally for 30 days) and kidney-yang-deficient model group (400 mg/kg hydroxyurea tablets at 8:00 a.m. daily, then treated with 80, 160, 320 mg/kg orally, daily for 18 days). In both groups, testosterone level was increased and cortisol, SOD, GPx, CAT, MAD levels were decreased. These results showed that bajijiasu enhanced the sexual behavior of both normal and kidney-yang-deficient mice. This study demonstrated that *Morinda officinalis* F. C. could be a potent natural androgen-like drug to enhance sexual function and prevent the testicular toxicity induced by environmental toxicants. Soheir et al. reported that *Balanites aegyptiaca* would be a good natural source for protective against male infertility [48]. The extract and toxic material were fed with daily food. Administration of *Balanites aegyptiaca* sapogenin extract (25, 50, 100 mg/kg for 70 days) to albino rats showed a curative effect against 34 mg/kg of AlCl_3_ induced infertility and dysfunction, increasing LH, estradiol, testosterone, and glucose levels, at the same time, decreasing FSH, cholesterol, sAST, urea, creatinine levels. Additionally, the semen quality was improved in sperm motility, sperm count, and sperm transit rate. Extract dose at 100 mg/kg showed the highest protective effect. This study indicated that *Balanites aegyptiaca* possessed protective effects against male infertility induced by aluminum chloride. Zakaria et al. reported that bee bread supplementation produced from bee pollen may cause a positive effect on sperm count and sperm morphology [49]. SD rats were administered with 0.5 g/kg body weight of bee bread (diluted in 1 mL of distilled water) by oral gavage once daily for 28 days. Supplementation of bee bread significantly increased the weight of prostate glands. Sperm count and sperm morphology did not show significant differences compared to the control group but there was an increasing pattern. This study showed bee bread supplementation may cause a positive effect on sperm count and sperm morphology. *Cistanche tubulosa* (Schrenk) R. wight is a phanerogamic and perennial plant from the Orobanchaceae family which is used to cure reproductive dysfunction and enhance male sexual function [50]. *Cistanche tubulosa* extract (200 mg/kg for 42 days) and echinacoside at a dose of 150 mg/kg for 30 days attenuated testicular toxicity induced by bisphenol A in SD rats. Both products showed remarkable recovery in daily sperm production, abnormal morphology (isolated head, head without curvature), and immobile sperm. Hormone levels (LDH-x activity and testosterone) and steroidogenic enzyme activities (3β-HSD, 17β-HSD, CYP17A1, CYP11A1, and StAR) were upregulated in both products. Meanwhile, the number of sperm in testis and levels of LH and FSH were significantly changed only by the administration of echinacoside. These findings concluded that the natural products inhibited the deleterious effects of bisphenol A, and protected sperm and testis injury. Yari et al. elucidated the profertility effects of aqueous extract of *Crocus sativus* Linn., which is an anticatarrhal, nerve sedative, and aphrodisiac medicine commonly known as saffron [27]. The administration of the extract to cadmium-exposed Wistar rats resulted in an increase of mean sperm number and a regulation of testosterone and LPO levels. Histological findings showed increased cell proliferation, which indicated that *Crocus sativus* Linn. could protect cells from oxidative stress. Oguzturk et al. reported that curcumin found in turmeric may be useful for the prevention of CdCl2 induced reproductive damage [51]. CdCl2 dissolved in distilled water (1 mg/kg, intraperitoneally) and curcumin suspended in corn oil (100 mg/kg, orally) were given together for 3 days. SD rats administered significantly decreased TBARS level and increased GSH, CAT, SOD, and GPx activities. This study demonstrated that curcumin treatment ameliorated the toxic effects of CdCl2 in the light of lipid peroxidation by biochemical and histopathological alterations. Eva Tvrdá et al. reported that curcumin derived from *Curcuma longa* Linn. seemed to protect spermatozoa against the damage caused by the hostile in vitro culture [52]. Bovine cells were cultivated with curcumin (1, 5, 10, 50, 100 μM/L) for 2, 6, 12, 24 h. The administration increased spermatozoa, increasing metabolic activity and decreasing nitroblue-tetrazolium chloride. It showed highest efficacy in the concentrations of 10 and 50 μM/L. This finding concluded that *Curcumin longa* Linn. had an ameliorating effect on male fertility. Rahim et al. reported that *Cymbopogon citrates* effectively protected reproductive organs [4]. SD rats were given 100 mg/kg *Cymbopogon citrates* aqueous extract and 0.5% H_2_O_2_ in addition to their standard food and drink for 30 days. After administration GSH level increased, MAD level decreased. In addition, the body, testicular, and epididymal weight, and testosterone level were increased. Additionally, oxidative stress and protected male rats against H_2_O_2_-induced reproductive system injury were reduced. This study showed that *Cymbopogon citrates* possessed a protective effect on H_2_O_2-_induced oxidative stress in the reproductive system of male rats. El-Kashlan et al. reported that *Phoenix dactylifera* L. also named as date palm showed alleviation of testicular damage in Wistar mice with thyroid disorders [53]. Extract from this compound (150 mg/kg for 56 days) elevated the levels of LH, FSH, and testosterone, while it downregulated the level of estradiol. This was accompanied by the active expression of 3β-HSD and 17β-HSD enzymes. These results elicited that date palm pollen extract had potential protective effects against testicular dysfunction induced by thyroid hormones. Khaki et al. reported that diallyl sulfide isolated from *Allium sativum* possessed a potential therapeutic effect against lead(Pb)-induced testicular toxicity and its underlying mechanisms [54]. Albino rats were treated with PbAc (20 mg/kg), dissolved in distilled water, 30 min before diallyl sulfide (200 mg/kg) dissolved in corn oil (1.0 mL/kg). All of the administrations were done daily and orally for 49 days. It elevated the SOD and GSH levels, increasing the number of sperms, weights of testes and epididymis, and serum testosterone. Diallyl sulfide seemed to be a promising agent for protection against lead (Pb)-induced testicular toxicity inducing material through antioxidative properties, beside regulation of testicular apoptosis and aromatase expression. Fakher et al. reported that *Echium amoenum* had a potential ameliorative role in male reproductive parameters and positive impact on boosting fertility [34]. Mus musculus mice received borage distillate of 1/2 dilution (150 ± 2.5 mL/kg) or 1/4 dilution (75 ± 1.25 mL/kg) for 3 weeks and distilled water for 2 weeks, 5 weeks in total, which was the length of spermatogenesis in mice. The administration increased serum FSH, LH, testosterone, sperm motility, and the number of Leydig cells. These results showed that *Echium amoenum* could improve the male reproductive hormones and parameters. Arafa et al. reported that *Echinacea purpurea* possessed significant improvement in the antioxidant status of the testicular tissue against cyproterone acetate [26]. *Echinacea* extract (63 mg/kg) and cyproterone acetate (25 mg/kg) were given to the Wistar strain rats for 1 week before the combination treatment. After 3.5 h cyproterone acetate administration as the peak plasma concentration, it was followed by *Echinacea* extract treatment. Administration was conducted daily via an oral tube for two intervals for 2 or 4 weeks. It increased SOD, GST, sperm count, calcium ion contents while decreasing MDA, nitric oxide (NO) levels. This study showed that *Echinacea purpurea* had antioxidant effects against stress induced by cyproterone acetate treatment in the male productive system. Echinacoside is derived from *Cistanche tubulosa* Hook f., a traditional Chinese medicine utilized to cure reproductive dysfunction and boost male sexual activity [55]. Jiang et al. demonstrated that echinacoside at doses of 5, 20, 80 mg/kg for 14 days improved sperm count in cauda epididymis and elevated LH, CYP11A1, CYP17A1, HSD3β1/2, HSD17β, and StAR, when treated to Kunming mice. The restriction of AR activities was estimated by the upregulation of various hormone related genes (LHβ, LHr, Gnrh 1, and Gnrhr) and the reduction of hypothalamic AR translocation to the nucleus. Furthermore, echinacoside was effective in bisphenoal A induced mice, by mediating endocrine related protein activation (CYP11A1, CYP17A1, HSD17β, and StAR) and improving sperm count, and motility [50]. These findings supported that echinacoside was able to regulate a wide range of fertility related mechanisms, which protected oligoasthemospermia in mice. Mozafari et al. reported that ethyl pyruvate had a protective effect on sperm quality parameters, testosterone level, and malondialdehyde in phenylhydrazine treated mice [29]. NMRI mice were treated with initial 8 mg/100 g phenylhydrazine followed by 6 mg/100 g, every 48 h by intraperitoneal injection. Additionally, they received 40 mg/kg ethyl pyruvate daily for 35 days by intraperitoneal injection. Administration of ethyl pyruvate improved viability power of sperm, immaturity of sperm nucleus, sperm mobility, and morphology a little. MDA level was decreased. This study showed that ethyl pyruvate as an antioxidant could reduce destructive effects of phenylhydrazine on sperm parameters, testosterone level, and lipid peroxidation. Wahab et al. reported that *Eurycoma longifolia* heightened testicular functioning, reversing the effects of estrogen by increasing spermatogenesis and sperm counts [56]. SD rats were treated with 8 mg/kg body weight of *Eurycoma longifolia* extract orally with an intramuscular injection of 500 mg/kg body weight estradiol. *Eurycoma longifolia* was dissolved in normal saline and estradiol was diluted using ethanol and olive oil (1:9 volume). All treatments were administered daily at 09:00 am and were continued for 14 days. The extract heightened testicular functioning and inhibited the effects of an excessive estrogen state which could result in germ cell apoptosis. Additionally, it enhanced sperm motility and exhibited higher sperm counts when compared to the estradiol group. Therefore, *Eurycoma longifolia* can be used as a supplement for treating fertility conditions where there was testosterone deficiency or estrogen excess. *Ginkgo biloba* Linn. is known to contain numerous natural substances which were reported to have antioxidant, antiallergic, anti-inflammatory, antiproliferative, and anticarcinogenic effects [57]. Wistar rats were induced testicular damage by ischemia/reperfusion injury in order to mimic testicular torsion, and were observed after treatment with *Ginkgo biloba* extract (50 mg/kg for one time). It was indicated that the extract increased seminiferous tubular diameter, along with the number of primary spermatocytes, round spermatids, and Leydig cells. The levels of testosterone, FSH, mitochondrial NAD, plasma TNF-α, and IL-1β were regulated back to normal. Histopathological findings showed *Ginkgo biloba* attenuated degeneration of germinal epithelial lining, apoptosis of germinal cells, and abnormalities in sperm, which further supported its profertility effects. Khaki et al. reported that *Zingiber officinale* Rhizoma had a useful effect on spermatogenesis [58]. Wistar rats received ginger powder at doses of 50, 100 mg/kg daily for 20 days and after treatment, TAC level increased, MDA level decreased. Especially, the dose at 100 mg/kg significantly increased testosterone, sperm viability, and motility, preserving LH, FSH hormones, sperm concentration, morphology, and testes weights. This result suggested that *Zingiber officinale* Rhizoma may be promising in enhancing sperm healthy parameters. Gang et al. reported that grape seed from *Vitis vinifera* alleviated arsenic-induced pathological changes and oxidative stress damage in mouse testis [59]. Kunming mice were administered 4 mg/kg arsenic trioxide (ATO) in the morning and grape seed proanthocyanidin extract (100, 200, 400 mg/kg) in the evening daily by gavage for 5 weeks. The extract treatment increased Nrf2 signaling and GST, HO1, NQO1, SOD, T-AOC levels and decreased MDA and 8-OHdG. Additionally, it showed increased sperm count and lower rate of sperm abnormalities. This finding concluded that *Vitis vinifera* possessed protective effects against reproductive toxicity. *Ionidium suffruticosum* (L.) Ging (Violaceae) is an ethnomedicinal herb widely used for treating jaundice, male sterility, diabetes, malaria, urinary tract infections, and water retention [60]. Carbendazim induced subfertile albino rats, treated with methanol extracts of *Ionidium suffruticosum*, showed an increase of sperm count, cauda epididymis sperm motility, body and testis weight, and germinal epithelial cell mass. It also showed decreases of epidermal sperm agglutination, and abnormalities of sperm morphology (detached tail, fusion of sperm, broken middle piece, detached tail, coiling of flagellum). Histopathological results confirmed improvements in arrangements in seminiferous tubules and germinal epithelium, showing all stages of meiotic division. The level of CAT, SOD, and MDA activities were brought back to normal, further indicating the reproductive enhancing effects of this natural compound. *Jurenia dolomiaea* Boiss., one of the Asteraceae family is referred to as a prostrate perennial herb that is used for antiseptic skin eruptions, stomach ache, and loose bowels [61]. Carbon tetrachloride (CCl4) induced SD mice were treated with methanol extracts (200, 400 mg/kg for 60 days) of this natural product, and the results showed an augmentation of antioxidant enzymes like SOD, CAT, POD, GST, GPx, and GR, and hormones such as testosterone and GSH. Concentration of H_2_O_2_, TBARS, and nitrate were lowered, and thicknesses in germinal layers were also recovered in histological observation, showing antioxidative and reproductive properties of *Jurenia dolomiaea* Boiss. Park et al. reported that KH-465, which is a mixture of herbal extracts from *Epimedium koreanum* Nakai and *Angelica gigas* Nakai had a protective effect on spermatogenesis [5]. SD rats received LH releasing hormone agonist for 4 weeks to induce spermatogenic failure and 200, 400 mg/kg of KH-465 were administered. KH-465 was dissolved in distilled water and administered orally through an 8F red Rob-Nel catheter. This treatment increased LH, SOD levels, while decreasing 8-OHdG level. In addition, it increased the sperm count and motility. These results suggested that KH-465 increased sperm production and had a positive effect in a male infertility model. Ebokaiwe et al. elicited that *Loranthus micranthus* Linn., a traditional medicine used for a broad range of diseases such as hypertension and cancer, had ameliorative effects on hyperglycemia mediated testicular dysfunction [62]. The aqueous methanol extracts from this natural compound (100, 200 mg/kg for 14 days) were administrated to Wistar rats which were rendered by streptozotocin. It was observed that testis weight and various sperm parameters were regulated back to normal. These outcomes were related to the increase of testosterone, FSH, LH, and steroidogenic enzyme activities (3β-HSD and 17β-HSD). There was also elevation of SOD, CAT, GSH, GSH-Px, GST, and decreases in MDA, LPO, and Bcl-2. These results confirmed the protective prospects against glycemia-related reproductive disruption. *Lycium barbarum* Linn. is a well-known Chinese medicinal herb used for nourishing liver and kidney, and improving eyesight [63]. Shi et al. reported that Institute of Cancer Research mice injected with streptozotocin, when treated with this natural compound, showed improvement of testis weight, epididymis weight, sperm viability, and the recovery of spermatogonia and Sertoli cells. Furthermore, treated with *Lycium barbarum* polysaccharide (20, 40 mg/kg for 62 days), mating behaviors such as mounting, intromission, and ejaculatory latency with post-ejaculatory intervals were rated back to normal. Additionally, sex hormone levels (testosterone, FSH, and LH) and blood glucose levels were also regulated. These data revealed the promising role of *Lycium barbarum* as a protective agent for fertility damage caused by endoplasmic reticulum stress and diabetes. *Lepidium meyenii* Walp. is a Peruvian hypototyl and its roots are widely used for energizing and enhancing fertility [64]. Maca capsules (500, 1000 mg/kg for 28 days) were administrated to Empire Breeders mice induced with cyclophosphamide. Various sperm parameters (sperm count, sperm motility, seminiferous tubule width, germinal cell layer thickness), endocrine activities (testosterone and GSH), and antioxidant protein expressions (CAT, SOD, and MDA) were reversed back to normal. These results indicated that Maca attenuated chemical infertility through regulation of cytochrome p450 and androgen production, which was most distinct at the maximum dose of 100 mg/kg. Cuya et al. also investigated the effects of Maca extract (666 mg/kg for 35 days) on chemically and physically subfertile BALB/c mice [65]. Oral administration of ketoconazole and exposure of magnetic fields were utilized to render subfertility, and treatment with Maca extracts increased sperm motility and sperm count. DNA fragmentation decreased in the physically subfertile group, yet the differences were not significant. Afrigan et al. reported that *Matricaria chamomilla* hydroethanolic extract had a protective effect on adverse effects of formaldehyde on the male reproductive system [25]. Wistar rats were injected with 10 mg/kg of formaldehyde and 200, 500 mg/kg of extract intraperitoneally and daily for 30 days. The extract increased testosterone and LH level, enhancing sperm count, motility, and viability. Decrease was evident in the number of apoptotic germ cells at dose of 500 mg/kg. These results suggested that *Matricaria chamomilla* had ameliorative effect on male infertility. Saleem et al. reported that *Mentha Spicata* especially, at dose 400 mg/kg exhibited a strong antimutagenic effect against ifosfamide clastogenic action in bone marrow cells and sperm abnormalities [36]. In vivo, Mus musculus strain BALB/c were treated with 50 mg/kg Ifosfamide intraperitoneally at the beginning of the week for one time and also were treated with *Mentha spicata* aqueous extract (40, 100, 400 mg/kg) for one week. This treatment showed significant decrease in abnormal sperm, at the same time decreasing the effect of ifosfamide on sperm morphology. This result showed that *Mentha spicata* had a protective effect on sperm abnormalities induced by infertility inducing toxic chemicals in male productive systems. Sahreen et al. reported that *Carissa opaca* leaves methanolic extract had a therapeutic effect against CCl_4_ induced reproductive abnormalities [7]. Administration of CCl_4_ (0.5 mL/kg b.w., 20%/olive oil) was intraperitoneally twice a week for 8 weeks. At the same time and for the same duration, the rats were administered silymarin (50 mg/kg b.w.) and extract (100, 200 mg/kg b.w.) orally. SD rats treated with extract showed an ameliorating effect against testicular toxicity provoked by CCl_4_. SOD, CAT, POD, GPx, GST, GR, and QR activities improved while triglycerides level, total cholesterol, HDL, LDL, thiobarbituric acid reactive substances (TBARS), and H_2_O_2_ levels were decreased. Treatment with silymarin also produced similar results. It could be concluded from the current study that bioactive components of MLC, especially flavonoids (myricetin, kaempherol, isoquercetin, hyperoside, and vitexin), have the ability to recover the metabolic enzymatic activities and repair cellular injuries, thus providing scientific evidence in favor of its pharmacological use in testicular dysfunction. This finding suggested that *Carissa opaca* leaves possessed a curative effect to male infertility. Zade et al. demonstrated that *Moringa oleifera* Seed enhanced sexual behavior in male rats [66]. Wistar strain albino rats were administered with *Moringa oleifera* Seed aqueous extract at doses of 1000, 2000, 5000 mg/kg orally for 7 days in set I and 100, 200, 500 mg/kg orally for 21 days at 18:00 h in set II. In set I, conducted for acute toxicity study, healthy male albino rats were starved for 3–4 h and subjected to acute toxicity studies. The extract was devoid of any adverse effects and acute toxicity. Additionally, in set II, conducted for reproductive ability test, after treatment, there was an increase in the mounting frequency, intromission frequency, ejaculatory latency, and libido and mating activities. The most effective dose was 500 mg/kg. These results clearly indicated that *Moringa oleifera* had a positive effect on spermatogenesis in male rats. *Moringa oleifera* Lam. is a traditional medicine used in Nigeria and is known to have positive effects on sexual behavior and reproductive function [17]. Leaf powder from this natural compound (5, 10, 15 g/kg for 12 weeks) administered to New Zealand White rabbits elevated semen volume, sperm count, and motility, while it reduced abnormal morphology of sperm. These morphological changes and the increase of FSH and LH levels all together elucidated the profertility features of this natural compound. Takayev et al. also reported the effects of extracts from *Moringa oleifera* Lam. (400, 800 mg/kg for 2 weeks) treated to cryptorchidism induced SD rats. The results revealed increases in germinal cell layer thickness, diameter of seminiferous tubules, testis weight index, and testicular weight, but decreases in perivascular fibrosis and apoptotic cell detection [67]. The morphological changes were observed to be related to increased expressions of SOD and lowered levels of HSP70 and MDA. These findings confirmed the effects of *Moringa oleifera* Lam. ameliorating testicular impairments and bioactive changes due to oxidative stress in male reproductive organs. El-Wakf et al. reported that marjoram originated from *O. Marjoram* and Sage isolated from *S. officinalis* seemed to prevent reproductive disorders caused by obesity, recognized as a leading cause for male infertility [23]. In set I Wistar albino rats were fed high fat diet and received orally marjoram oil (0.16 mL/kg b.w. daily for 12 weeks) diluted in sunflower oil (1:2) and in set II sage oil in the same way and doses. Administration of marjoram or sage oil extracts to obese animals reduced testicular lipid accumulation and elevated androgens and sperm count, in addition to improving testicular structure. This report suggested that both types of oil should be considered in future therapeutic approaches for controlling adverse impacts of obesity on male fertility. Adana et al. assessed the beneficial effects of naringenin (40, 80 mg/kg for 10 weeks) to treat reproductive problems in highly active antiretroviral therapy (HAART) exposed SD rats [68]. Naringenin is a flavone extract from citrus species which is famous for its bioactivities related with scavenging free radicals and protecting testicular toxicity. Increase of progressive motility and restoration of seminiferous tubules and epithelium volume fraction were observed. Although changes in sex hormone levels (testosterone and LH) were insignificant, histo-morphology displayed larger seminiferous tubules and improvements in the number of Leydig cells and sperm cells. These data suggested the beneficial prospects of narginen against HAART testicular and reproductive damage. Haseena S et al. reported that *Nigella sativa* seed powder improved the reproductive efficiency [69]. The albino rats were given 50 mg/kg streptozotocine intraperitoneal injection to induce diabetes and their testosterone and LH levels were decreased in diabetes state. In *Nigella Sativa* seed powder (300 mg/kg orally for 45 days) treated rats, the testosterone levels were increased significantly. Hence the study suggested that the *Nigella sativa* seed may be used for increasing testicular activity. Pedalium murex methanol fruit fraction originated from *Pedalium murex* Linn. which is a famous folk medicine used for treating ulcers, digestive tonics, and puerperal diseases [70]. Sulphasalazine induced rats treated with this natural compound showed improved sperm motility, sperm density, spermatogenesis, and the number of germinal cells, interstitial cells, spermatid, preleptotene spermatocyte, fibroblast, and mature Leydig cells. The levels of LH, FSH, testosterone, cholesterol, glycogen, and sialic acid came back to normal, which supported the fertility enhancing activities. *Petasites japonicus* MeOH extract proliferated the generation and activity of spermatogonial stem cells when treated to SSC/C57BL/6 mice at doses of 0.1, 1, 10 μg/mL for 7 days [71]. Elevated levels of LHX1 and GFRα1 were observed, which were used as markers of differentiation of spermatogonia at the basement membrane of seminiferous tubules. These data elicited that buthanol fraction of *Petasites japonicus* had the capability to preserve normal spermatogenesis. S. Bahmanpour et al. reported that *Phoenix dactylifera* date palm seemed to cure the male infertility by improving sperm quality [72]. SD rats were administrated with *Phoenix Dactylifera* pollen (30, 60, 120, 240 mg/kg in distilled water) orally for 35 consecutive days. The most effective dose was 120 mg/kg. It increased sperm count, testosterone level, and the number of Leydig cells by 30% while reducing DNA denaturation. These results showed *Phoenix Dactylifera* date palm pollen suspension could enhance male fertility. *Phyllanthus emblica* Linn. is a plant which contains phenolic compounds and exerts ROS scavenging capacities [73]. SD rats suffered from chronic stress induced through immobilization, but showed reversing tendencies in sperm concentration, testicular size, sperm head abnormality, and acrosome-reacted sperm, when treated with the extract of this natural product (50 mg/kg for 42 days). These preventive effects were explained by the improvement of testosterone level, StAR protein, and tyrosine-phosphorylated proteins of 60, 51, 23 kDa, and a reduction in levels of corticosterone, MDA, and tyrosine-phosphorylated proteins of 14, 34 kDa. This demonstrated that *Phyllanthus emblica* Linn. attenuated testicular damage especially through the control of tyrosine-phosphorylated proteins. *Pilea microphylla* (L.) Liebm. is a folk medicine used as a cure for several allergies and has been reported to have antioxidant activities [74]. It was observed that varicocelized Wistar mice treated with its extract (50 mg/kg for 10 weeks) showed increased left epididymal sperm count, motility, vitality, and the number of sperm with normal morphology. Meanwhile, chromatin staining assessment revealed that *Pilea microphylla* did not improve sperm nuclear integrity through chromatin condensation. *Pistia stratiotes* Linn. which is known as water lettuce has been ascribed to exert anthelmintic, antimicrobial, and antifungal effects [75]. The ethanol extract (100 mg/kg for 14 days) treated to arsenic-induced Wistar strain albino rats, showed restoration in sperm motility and abnormality. The results with sperm viability, sperm volume, and sperm count were not significant, yet the tendency showed promising effects of *Pistia stratiotes*. *Rosa damascena* Linn., originated from the Rosaceae family are utilized for various conditions such as loss of libido, Alzheimer’s disease, cardiovascular disorders, and diabetes [33]. Askaripour et al. reported the antioxidative potential of *Rosa damascena* aqueous extract (10, 20, 40 mg/kg for 40 days) on formaldehyde induced testicular damage in NMRI mice. The extract restored sperm and testis parameters (sperm number, motility, viability, rate of normal sperm, testis weight, length, and width), the number of Leydig cells, along with testosterone levels. Histopathological changes showed an increase in the diameters of seminiferous and epithelial tubules, further identifying the protective effects of this natural compound. *Chlorophytum borivilianum* Santapau and Fernandes is a natural product used as a nutritive tonic for sexual weakness and is also called as safe musil [76]. Its extracts (125, 250 mg/kg for 52 days) showed increases of sperm count and mount latency when treated to Wistar mice which were previously administrated with sildenatil citrate. Although the changes in sperm abnormality were insignificant, the results all together supported safety and aphrodisiac properties of safe musil against infertile mice. Luthfi et al. reported that *Lunasia amara* possessed a positive effect on spermatogenesis [77]. SD rats were given *Lunasia amara* aqueous extract at doses of 30, 60, 90 mg/kg by force-feeding between 10:00 am and 12:00 pm daily for 42 days. It increased the sperm count, progressive motility, and seminiferous tubules diameter. This study concluded that *Lunasia amara* was a potential herb to enhance male fertility. Mishra et al. demonstrated the profertility effects of Shilajit which is a compact humic substance commonly prescribed for genitourinary disorders, digestive disorders, and diabetes [78]. Toxicity induced by cadmium in Swiss albino mice was reverted after treatment with the water extract (50, 100, 200 mg/kg for 35 days). The results showed increase in weight of testis, epididymis, seminal vesicles, and sperm parameters (sperm production, concentration, and motility), and a decrease of affected seminiferous tubules. Concentration of sialic acid in epididymis, and fructose in seminal vesicle was elevated along with higher levels of libido and male fertility index. Abedi et al. reported that silymarin from *Silybum marianum* seed increased testosterone hormones [79]. Wistar rats were treated with silymarin at concentrations of 50, 100, 150 mg/kg by gavage for 28 days. This administration increased the number of spermatids and spermatozoa cells in rats. It also increased the secretion of LH, FSH, GnRH. Dose at 150 mg/kg showed a significant effect. This report showed that silymarin could contribute to increase of testosterone secretion and improvement of the spermatogenesis process. Kanedi et al. reported that Suruhan which is from *Peperomia pellucid* possessed positive effects on blood glucose levels and sperm parameters in male albino mice with alloxan-induced hyperglycemia [80]. Injection of alloxan inducing hyperglycemia was done three times in 6 days at the dose of 150 mg/kg. Administration of suruhan ethanolic extract (56, 112, 168 mg/kg for 35 days) to mice significantly lowered blood glucose levels, ameliorated sperm count, viability, motility, and morphology. This study concluded that *Peperomia pellucid* had the potential as antidiabetic and male fertility recovery agents. Akomolafe et al. reported that *Tetracarpidium conophorum* leaf extract, known for its pharmacological effects to treat eczema, pruritus, psoriasis, and prostate cancer, alleviated toxicity induced by ethanol on reproductive organs and semen quality [81]. After treatment (500, 1000 mg/kg for 21 days), ethanol induced Wistar rats showed recovery of testis weight, sperm concentration, viability, motility, normal/abnormal chromatin integrity, and total sperm abnormality (tailless head, headless tail, bent tail). The results revealed an upregulation of G6PD, 3β/17β-HSD activity, testicular glycogen, Zn and Se content, epididymal Zn and Se content, as well as sex hormones (FSH, LH, and testosterone). Meanwhile, the level of testicular cholesterol was downregulated, further identifying that *Tetracarpidium conophorum* could activate steroidogenesis and protective effects on reproduction. *Tetracarpidium conophorum* (Müll. Arg.), also known as walnut leaf, is rich in zinc and vitamin C and has been reported to have effects on sperm quality and endocrine levels [82]. Akomolfe et al. identified that ethanol induced Wistar albino mice treated with walnut leaf extracts (500, 1000 mg/kg for 21 days) enhanced sperm parameters such as testis and epididymis weight, sperm count, sperm motility, curvilinear velocity, and epididymal sperm viability. Biochemical parameters such as G-6PDH, LDH, 3β-HSD, 17β-HSD, testicular Zn and Se content in testis and epididymis, as well as testicular glycogen and cholesterol content were reversed back to normal. The efficacy of this extract was proved by better results in sex hormone levels and 17β-HSD activity, when compared to clomiphene citrate, a commonly used fertility drug. *Teucrium polium* Linn., one of the wild growing flowering species is famous for its antioxidant, anti-inflammatory, antirheumatic, and hypoglycemic effects [32]. Rahmouni et al. demonstrated that *Teucrium polium* extract (1 mL/kg for 10 weeks) elevated sperm motility, sperm count, and lowered sperm abnormality in carbon tetrachloride (CCl4) induced Wistar rats. It was also discovered that the alteration of levels of sex hormones (testosterone, FSH, LH), antioxidant enzyme GPx, TBRAS, and antioxidant enzymes such as CAT and SOD were alleviated. These findings suggested the effects of this natural compound as an antioxidant and protective agent against chemically induced reprotoxicity. Geusmi et al. demonstrated the efficacy of *Thymus algeriensis* extract (150 mg/kg for 2 weeks) against H_2_O_2_ induced toxicity in Wistar rats [83]. *Thymus algeriensis* Lam. is a well-known herb famous for its anti-inflammatory, antitussive, antimicrobial, antiparasitic, and antiulcer effects. Given to rats with oxidative stress, the extract elevated sperm count, viability, motility, and the percentage of normal morphology. The degradation of antioxidative enzymes (CAT, SOD, GPx, and GST) and GSH were prevented. Decreased level of LPO was also observed, supporting improvements of fertility actions by this natural compound. *Tribulus terrestris* Linn., a member of the Zygophyllaceace family is commonly used as a folk medicine for a broad range of diseases from impotence, rheumatism, hypertension, to kidney stones [9]. Pavin et al. showed that *Tribulus terrestris* dry extract (11 mg/kg for 14 days) protected infertility in Swiss albino mice induced by cyclophosphamide, a famous anticancer drug. It was identified that *Tribulus terrestris* enhanced sperm motility while it restored the level of antioxidant enzymes (SOD, CAT, GPx, and GST), RS, TBRAS, as well as sex hormone related parameters like testosterone and 17β-HSD. Oliveira et al. discussed the efficacy of white tea, which is known to have high antioxidant properties [84]. Ingestion of white tea (1 g/100 mL for 2 months) on STZ induced prediabetic Wistar rats regulated testicular antioxidant potential and oxidative stress in testicular tissue. It was further elucidated that this natural compound was capable of regulating diabetic conditions through improving insulin sensitivity and glucose tolerance while reducing insulin resistance. Sperm concentration, viability, motility and morphology were modulated, which were investigated to be correlated with the changed values of the FRAP and testis carbonyl content. These results demonstrated that white tea counteracted the deleterious effects of male reproductive function caused by diabetes. *Angelica keiskei* Koidz. is an indigenous plant of Japan and its major polyphenolic compounds are known to exert antihypertensive, antiobesity, and antidiabetic effects [1]. Kokubu et al. analyzed the preventive role of *Angelica keiskei* powder (57.5 mg/kg for 7 days) and its functional component xanthoangelol (3 mg/kg for 7 days) when treated to heat stress induced sperm in self-breeding CD-1 mice. Both substances displayed positive outcomes upon sperm motility, progressive sperm density, progressive sperm velocity, seminiferous tubule abnormality, and the level of glutathione synthetase (GSS). However, *Angelica keiskei* powder showed better efficacy in improving the density of sperm and relieving DNA fragmentation, while it also augmented the expression of HO-1, Hspa11, Hspa2 Hsf1, Hsf2, and heme oxygenase-1. It was concluded that these phytochemicals possessed efficacy in preventing heat-stress induced male infertility. Oridupa et al. reported that *Xanthosoma sagittifolium* Linn., a natural diabetic remedy, prevented deterioration of reproductive organs induced by alloxan [85]. Experimentally induced diabetic Wistar mice were treated with *Xanthosoma sagittifolium* extract (25%, 50%, 75%, or 100%, 100 mg/kg for 14 days) and showed increases in sperm count, motility, and livability, and a decrease of luminal diameter. Histology of testes displayed reduction in seminiferous tubule diameter and luminal diameter, which indicated the possibility of the natural compound as a source to enhance male fertility. Akomolafe et al. reported that leaf extract from *Tetracarpidium conophorum* (Mull. Arg.) Hutch. and Dalz, commonly used as a fertility enhancing drug, had antioxidant properties in ethanol induced reproductive toxicity [86]. Treated with the extract for 21 days at doses of 50, 500, 1000 mg/kg, Wistar rats showed significant levels of recovery in the levels of MDA, GSH, and vitamin C. Improvements in degenerated testis lumen was observed, further demonstrating the ameliorative effects of *Tetracarpidium conophorum* on reproductive testis damage caused by radical species. Saeid et al. reported that *Zingiber officinale* had useful effects on spermatogenesis and sperm parameters in broiler breeder males [87]. The 24 week old broiler breeders were administered aqueous extract of *Zingiber officinale* in drinking water at 5% and 10% daily for 20 weeks. It increased FSH, testosterone, and LH levels, but decreased MDA and TAC levels. There were increases in testes weight, ejaculated volume, sperm concentration, and motility but decreases in dead sperm and abnormal sperm. These findings showed that *Zingiber officinale* could enhance spermatogenesis.

##### In Vitro and in Vivo Studies

Several studies were conducted both in vitro and in vivo to prove the effectiveness of natural substances of plant origin (Table 3). *Echinacea purpurea* Linn. also known as purple coneflower contains active compounds with antioxidative, anti-inflammatory, antibacterial, antiviral, and anticancer properties [88]. Mao et al. reported that *Echinacea purpurea* ethanol extract encapsulated with chitosan/silica nanoparticles (nano-EE) ameliorated diabetes related male infertility. Treatment in LC540 with this compound at a dose of 25 μg/mL for 24 h showed improved cell viability with decreased NO production. STZ induced Sprague Dawley (SD) rats restored its reproductive function, which was demonstrated by regulation of seminiferous tubules diameter, germinal cell layer thickness, area of seminiferous tubules and lumen, sperm motility, and sperm DNA integrity. Anti-inflammatory effects were determined by the decrease of TNF-α and IL-1β, along with antidiabetic effects which were shown by reduced blood glucose and plasma insulin levels, and improved plasma fibroblast growth factor 21 resistance. Kong et al. confirmed the antioxidant, anti-inflammatory, and steroidogenesis inducing effects of *Cistanche tubulosa*, a non-shlorophyllic parasitic plant widely used for morbid leucorrhea, chronic renal diseases, constipation, and impotence [22]. Echinacoside, originated from this natural compound was administered to LC-540 and TM3 Leydig cells (5, 10 μM) which were pretreated with advanced glycation end-products (ACEs) and showed better cell viability and lower superoxide anion. Especially LC-540 Leydig cells showed significant increases of StAR, CYP11A1, CYP17A1, HSD17β3 proteins and downregulations of RAGE, NF-*κB* protein expression, and H_2_O_2_ production. Moreover, when given to diabetes induced SD rats (160, 320 mg/kg for 6 weeks), it was elucidated that echinacoside increased sperm number, sperm motility, seminiferous tubule thickness, and decreased sperm abnormality. These results were explained by the upregulation of KiSS1, SIRT1, GPR54, and SOCS -3 mRNAs in the hypothalamus, as well as the regulation of antioxidative enzymes, inflammation parameters, and other hormones. The data showed the echinacoside could prevent damage in sperm, which supports its profertility capacity.

#### 2.1.2. Animal Derived Natural Products for Treatment of Male Infertility

Natural products from animal origin were also mentioned to have profertility effects upon males in various studies (Table 4). Seres et al. reported the androgenic activities of drone milk, which is derived from the hypopharyngeal and mandibular glands of *Apis mellifera* [89]. The administration of this natural product (110 mg/kg for 5, 10 days) to SD rats increased the weight of glans penis, seminal vesicle, and levator ani muscles as well as the level of plasma testosterone and the expression of Slap mRNA. Additionally, the combination of methyl palmitate and methyl oleate, which originated from a bioactive fraction of drone milk, showed similar increase in the weight of androgenic organs (glans penis, seminal vesicle, and levator ani muscles) as raw drone milk. These results demonstrated that drone milk and its chemical compounds exerted androgenic effects justifying its traditional usage to treat sexual problems. Syazana et al. reported that several sperm parameters were positively affected by gelam honey obtained from *Apis mellifera* [21]. SD rats were force-fed daily with 1.0 mL/100 g normal saline (0.9%) and gelam honey for 60 days. After treatment, there were no significant differences for weight, length, and width of testis but, sperm count was significantly increased. Additionally, lower percentage of abnormal sperm heads and tails were observed. This report suggested that gelam honey has the potential to increase the fertility of male rats by increasing sperm count and number of sperm with normal morphology. Kumari et al. demonstrated that hydroethanolic extract of Indian propolis (400 mg/kg for 4 weeks) preserved fertility in Swiss albino mice which were treated with mitomycin C, a famous chemotherapeutic agent [90]. The natural compound was extract from propolis which is a complex mixture of various compounds collected by honeybees. It was observed that degradation of sperm parameters (sperm count, total motility, spermatozoa with normal head morphology, and spermatozoa with normal DNA, chromatin immaturity, and apoptosis in spermatogonial germ cells) and testicular damage (testis weight, number of tubules with complete spermatogenesis, diameter of seminiferous tubule, and the number of germ cells) were reversed. Testosterone and GSH levels increased and antioxidant enzymes (CAT and MDA) were restored, along with the expression of repair protein RAD51. These findings showed the potential of propolis extract to preserve fertility functions of males undergoing chemotherapy. Martinez et al. reported that spermaurin which was a purified 7.8kDa peptide from the venom of scorpion *Scorpio maurus palmatus* stimulated the motility of sperm from different mammalian species (cynomolgus monkeys, bovine, and mice) [91]. This work was an in vitro evaluation of the ability of spermaurin to improve sperm motility parameters. Bovine cells, incubated with F8 dilution 1/20 for 10 min and monkey’s sperm, treated with spermaurin for 10 min, showed increase of sperm motility. Oocytes of OFI mice, incubated with purified spermaurin dilution 1/40 for 4 h, increased in vitro fertilization (IVF) outcome. This study demonstrated that spermaurin could be used as a new tool for sperm motility enhancement.

#### 2.1.3. Fungus Derived Natural Products for Treatment of Male Infertility

One study was found to elucidate the efficacy in male fertility of fungus derived natural product (Table 5). *Antrodia cinnamomea* Chang. is a medicinal fungus which has been reported to possess anti-inflammatory, antioxidant, hepatoprotective, neuroprotective, and anticancer effects [92]. Johnshon et al. elicited that *Antrodia cinnamomea* ethanol extract (385, 770, 1540 mg/kg for 5 weeks) attenuated streptozotocin and nicotinamide induced diabetic toxicity in SD rats. It was elucidated that this natural product increased total sperm count and motility rate, and decreased abnormal sperm count and DNA damage in sperm. Endocrine activities (LH and testosterone) were elevated by higher activities of related proteins (StAR, CYP11A1, and 17β-HSD expressions). Oxidative stress (H_2_O_2_, MAD, and NO) were downregulated along with the regulation of SOD, and the reduction of ER stress. Lower expressions of GRP 78, and RAGE revealed that this natural product effectively controlled diabetic problems, which was correlated with reproductive disruption.

#### 2.1.4. Formula for Treatment of Male Infertility

Formula constituted with natural products was also noted to have positive effects upon male reproductive function (Table 6). Gosha-jinki-gan decoction is a formula of various natural substances which are potential agents against peripheral neuropathy and diabetic neuropathy [93]. Bulsulfan induced C57BL/6J mice treated with a diet including this formula (5.4% for 150 days) showed normalization of testis weight, epididymis spermatozoa count, fertility rate, and seminiferous tubule appearance. Additional findings revealed diminished levels of apoptosis which was observed by the decrease of caspase-8, F4/80, and Fas, as well as reduced macrophage migration by the suppression of TLR2 and TLR4 expression. This elucidated the possible usage of this formula to treat male infertility by preventing apoptosis of sperm. MOTILIPERM is a mixture of three natural extracts of *Morinda officinalis* How, *Allium cepa* L., and *Cuscuta chinensis* Lamark, which is used to treat rheumatoid arthritis and impotence [94]. Soni et al. reported that MOTILIPERM (100, 200 mg/kg for 4 weeks) exerted protective effects in varicocele induced SD rats by improving reproductive organ parameters (left testis weight, sperm motility, sperm count, epididymis motility, vas deferens count, epididymis count, spermatogenic cell density, and the percentage of damage of seminiferous tubule). The results also showed downregulations of MDA, testosterone, and ER stress-related molecules such as GRP-78, p-IRE1α, and p-JNK. Saikokaryukotsuboreito has been assessed to have numerous medicinal effects including depression, neurosis, anxiety, palpitation, vertigo, and insomnia [95]. Treated to aging C57BL/6 mice, Saikokaryukotsuboreito (300 mg/kg for 30 days) increased sperm number, sperm motility, testosterone in serum and testis, along with the SYCP3, which is a meiotic marker assessing the level of spermatogenesis. The results elicited that this mixture of various compounds showed benefits in spermatogenesis and fertility. Zhou et al. investigated that Shengjing capsule extracts, referred to as sperm-producing capsules, had antioxidant effects on cadmium chloride treated SPF Wistar mice [8]. The extracts (0.45, 0.90, 1.80 g/kg for 56 days) improved sperm motility, sperm morphology, and DNA fragmentation rate. Antioxidant enzymes like GSH-PX, MDA, and SOD, and testosterone were also observed to improve after treatment. Wuzi Yanzong pill, a Chinese polyherbal formula was identified to lessen testicular damage induced by X-rays in Kunming mice [30]. After treatment with this pill (1.0 g/kg for 21 days), testis weight, sperm count, and sperm motility increased with the upregulation of testosterone and total antioxidant status. Additionally, Wuzi Yanzong pill remarkably prevented alterations of PCNA and MDA, further supporting its antioxidative properties. These results demonstrated that various kinds of formula had the capacity to be utilized as effective agents against male reproductive dysfunction. Sheng et al. demonstrated that Yi Shen Jian Pi, a traditional Chinese medicine used to benefit “the kidney and spleen” exerted beneficial effects in semen quality and sperm mitochondria in oligoasthenozoosperma induced BALB/c mice [96]. It was determined that Yi Shen Jian Pi (1.35, 2.70 mg/kg for 4 weeks) augmented sperm quality, normal mitochondrial membrane potential, and the arrangements of fibers, axonemes, and mitochondrial sheath structures.

### 2.2. Natural Products That Reverse Female Infertility

#### 2.2.1. Plant Derived Natural Products for Treatment of Female Infertility

##### In Vitro Studies

Several in vitro studies showed that plant derived natural products possessed the capability to enhance fertility in females (Table 7). Nam et al. reported that *Evodiae Fructus* extract originated from *Evodia rutaecarpa* Benth [28]. When treated with 10, 50, 100, 300, 500 µg/mL for 24 h, it was reported that the extract protected ovary cells against 4-vinylcyclohexene diepoxide (VCD)-induced ovotoxicity via AKT activation, which was proved by the elevation of mTOR, and GSK-3β. Kim et al. reported that *Perilla frutescens*, which has been used to prevent threatened abortion in traditional medicine in the East Asian countries, demonstrated the effects on overcoming infertility due to implantation failure [15]. A total of 50 μg/mL of this natural compound administered to Ishikawa cell line and JAr cells for 48 h, and the increase of leukemia inhibitory factor (LIF) level, a major cytokine regulating endometrial receptivity, and LIF receptor in human endometrial Ishikawa cells was shown. *Perilla frutescens*-induced LIF expression increased the attachment of trophoblastic JAr cells to endometrial Ishikawa cells. This adhesion was mediated by the expression of integrin β5 and β3. These results suggest that *Perilla frutescens* enhanced the adhesion between Ishikawa cells and JAr cells and it can be a novel and effective candidate for improving pregnancy rate through the effects on reduced receptivity of endometrium, a cause of abortion.

##### In Vivo Studies

Numerous in vivo studies showed plant derived natural products possess the capability of enhancing fertility in females (Table 8). Ngoungoure Madeleine Chantal et al. demonstrated that *Anthocleista schweinfurthii* aqueous extract increased magnesium levels and exhibited antioxidative activity in postmenopause-like model of ovariectomized Wistar rats [97]. In total, 200, 300, 400 mg/kg of the supplement of this planet extract was treated to Wistar rats for 28 days, and it resulted in a considerable decline in malondialdehyde level associated with elevated Glutathione levels. Astragalus root, which is derived from *Astragalus mongholicus,* is shown to suppress estrogen-dependent endometrial hyperplasia and ovarian dysfunction [35]. Administration of this natural product to SPF/ICR mice 5% of powder diet for 56 days upregulated the expression of PPAR, mDECR, and increased blood estradiol level. El-Sayyad, Hassan I H, et al. demonstrated that barley (*Hordeum vulgare*) and date palm fruits (*Phoenix dactylifera*) protected the ovarian function from high cholesterol diet in *Rattus rattus* [98]. Supplement of those natural products, 10% of daily diet for 120 days, resulted in a considerable decline in MDA level associated with elevated CAT, SOD, and GST levels, which showed its antioxidative effects. Cinnamon powder, originated from *Cinnamomum verum*, has been reported to improve insulin resistance in PCOS [99]. The administration of cinnamon powder, at a dose of 10 mg/100 g for 20 days, increased the serum level of FSH, IGFBP-1 and decreased insulin, IGF-1 in DHEA induced PCOS mouse model. These findings demonstrated that cinnamon powder may be a potential therapeutic agent for the treatment of PCOS. Hee-Jung Choi et al. elicited that administration of *Cyperus rotundus* improved the adhesion of trophoblastic cells to endometrial cells for blastocyst implantation [12]. A total of 31.68 mg/kg of the water extract from this natural compound was administrated to C57BL/6 mice for 7 days. As a result, significant increase of LIF and LIF-dependent integrins αVβ3, αVβ5 was observed. These findings provide evidence that CR has therapeutic potential against poor endometrial receptivity Tiwari et al. reported that *Eucalyptus robusta* leaf extract inhibited the endometritis in Wistar rats at a dose of 25 mg/kg for 5 days [31]. It increased the level of COX-1 and COX-2 and decreased the level of TLR-4, TLR-9, MPO, iNOS, NO, and SAA. These findings showed that the natural compound could improve fertility through protecting against endometritis. Treatment with 10, 50, 100, 300, 500 µg/mL of *Evodiae Fructus* to CHO-K1 or COV434 cells for 24 h blocked the ovotoxicity induced by treatment with VCD, and significantly activated the Akt and its downstream effectors, such as mTOR and GSK-3β in CHO-K1 cells. These results demonstrated the potential utility of *Evodiae Fructus* as a natural remedy and alternative medicine for preventing premature ovarian failure which is a major cause of female infertility. Ngadjui E et al. reported that *Ficus asperifolia* aqueous extracts have alleviating effects on rat infertility induced by a high fat diet [100]. It decreased total plasma cholesterol, low-density lipoprotein (LDL) cholesterol and increased high-density lipoprotein (HDL) cholesterol in Wistar rats at a dose of 100 mg/kg once a day, 1 week in set I or 4 weeks in set II. This finding demonstrated that it has bioactive agents that may maintain conducive conditions for reproduction in some cases of infertility in obese women. Uchewa OO and Ezugworie 0OJ reported that *Ficus vogelii* aqueous extract had a protective role against lead induced reproductive toxicity [13]. The administration of this natural product at a dose of 100, 300 mg/kg for 21 days to Wistar rats decreased ovarian SOD. *Milicia excelsa*, a large deciduous fast-growing forest tree species native to tropical Africa, is widely used against female sterility, dysmenorrhea, and as aphrodisiac and galactagogue [18]. Mvondo, Marie Alfrede et al. demonstrated that the root aqueous extract of *Milicia excelsa* (14, 7, 140 mg/kg; 7, 15 days) may solve the problem of amenorrhea in Wistar rats by synchronizing the activity of the hypothalamic–pituitary axis to the ovarian production of estradiol and progesterone. It was also discovered that serum levels of FSH and estradiol were increased. These findings justify the traditional use of *Milicia excelsa* for primary and secondary amenorrhea. *Schisandra chinensis* is well recognized for its antioxidative, anti-inflammatory, and antidiabetic effects in traditional Chinese medicine [101]. The extract from this natural compound was administrated to Sprague Dawley rats at a dose of 200 mg/kg for 7 days. This led to the reduction of mRNA and protein levels of Prolactin in GH3 cells. These results show that *S. chinensis* and its single compound, gomisin N, may be candidates for treatment of hyperprolactinemia and prolactinoma. *Senecio biafrae* is a medicinal plant widely used by traditional healers in the western region of Cameroon for the treatment of female infertility [20]. Lienou et al. reported that *Senecio biafrae* (Oliv. and Hiern) J. Moore aqueous extract has puberty onset induction and ovarian folliculogenesis effect in immature female rats, and improvement of the various biochemical and physiological parameters of testicular germ cell apoptosis, fertility, light body weight gain variation, and increase in proteins, uterine weight was shown. In total, 8, 32, 64, and 128 mg/kg of this compound was treated to albino Wistar rats for 20 days, and increases of estrogenic activity, FSH, LH, estradiol, and progesterone levels was shown. Lastly, soy isoflavones, which originate from *Glycine max*, have been indicated to improve insulin sensitivity [101]. This natural product increased the production of SLRP in *Rattus norvegicus* Albinus at a dose of 150 mg/kg for 30 days. These findings supported the efficacy of this natural product to prevent insulin resistance, which is one of the factors that is related with infertility in females.

#### 2.2.2. Animal Derived Natural Products for Treatment of Female Infertility

A single study mentioned a natural product from an animal that showed the capacity to recover infertility problems in females (Table 9). Royal jelly is a dietary substance originated from *Apis mellifera* [102]. Elham Ghanbari et al. demonstrated that administration of royal jelly to Wistar rats (100, 200, 400 mg/kg for 14 days) resulted in the folliculogenesis by a significant increase of uterine and ovarian weights, the serum levels of progesterone, estradiol, FRAP, and a decrease in NO level.

### 2.3. Natural Products and Clinical Studies

Although clinical trials were scarce, there were studies based on human derived biological materials that revealed natural substances that were effective in treating infertility (Table 10). Wu et al. elucidated that bajijiasu isolated from *Morinda officinalis* F.C showed profertility properties against human sperm DNA which was obtained from 13 healthy donors [6]. Antioxidative effects were observed through the increase of CAT, GSH-Px, SOD, and the decrease of MDA. Ramen spectrum analysis identified that the data of 30% FeSO_4_/H_2_O_2_-damaged mitochondria of sperm were reversed back an average range after the administration of the extract for doses of 0.50, 1.00, 2.00 mg/mL for 45 min. This altogether demonstrated that bajijiasu could be used as a potent agent to treat oxidative stress related infertility matters. Inositol, found in natural sources such as wholemeal cereals, citrus fruits, brewer’s yeast, is known to regulate muscular, liver, and endocrine problems [103]. Seminal fluid obtained from healthy subjects and from subjects with oligoasthenoteratospermia were treated with inositol (2 mg/mL for 30 min, 1, and 2 h). Pathological samples treated with this natural compound showed improvement in spermatozoa intermediate tract thickness, as well as a reduction of amorphous materials and of damage in mitochondrial cristae. These data suggested that Inositol had a beneficial role in altering mitochondrial functions and fertility activities. Martinex et al. reported that spermaurin, a La 1-like peptide which was isolated from the venom of the scorpion *Scorpio maurus palmatus* improved sperm motility in fresh and frozen-thawed sperm of humans [91]. Human sperm, obtained from normozoospermic donors and subfertile patients, were treated with speramqurin and showed increased sperm motility. It was further demonstrated that this peptide was innocuous against sperm vitality, while the efficacy towards sperm motility was long lasting. A study on *Tribulus terrestris* Linn. was progressed by Khaleghi et al. to determine the effects of the extracts of the natural product based on semen samples from humans [104]. The extract (40, 50 mg/mL for 120 min) improved total sperm motility, the number of progressive motile spermatozoa, and curvilinear velocity. Asadmobini et al. also demonstrated the protective effects of this extract to improve sperm viability and motility [105]. Cryoprevented semen samples treated with *Tribulus terrestris* extract (20, 40, 50 μg/mL for 120 min) showed an increase of progressive motile spermatozoa, confirming the antioxidant properties of this compound. These studies showed that *Tribulus terrestris* Linn. is a promising tool to improve reproductive capacities in both mice and human semen. Chilean propolis ethanol extract originated from *Apis mellifera* has been reported to have antibiotic, antifungal, and anti-inflammatory effects [2]. This compound (6, 12, 25 μg/mL for 1 h) decreased the intracellular oxidants and DNA damage on normozoospermic semen samples treated with benzo(a)pyrene and exogenous ROS. TBARS and LDH, which were observed in sperm treated with ADP, H_2_O_2_, and FeSO_4_, showed significant downregulation patterns when cotreated with propolis extract. These findings demonstrated that extracts of propolis possess the potential to act as protective agents against oxidative products, which are one of the main causes of male infertility. Yue et al. reported that low-dose aspirin in combination with Tiao Jing Cu Yun pills increased the ovulation rate and the probability of pregnancy by promoting ovulation and increasing the thickness of the endometrium, thus creating a favorable internal environment for implantation [106]. It was given to PCOS patients twice per day until the day of ovulation at a dose of 100 mg per day, and effective improvement of perifollicular artery blood flow, and enhancement of the oocyte quality was shown. Kort et al. reported that cinnamon supplementation improves menstrual cycle [107]. A total of 45 women with PCOS were randomized to receive cinnamon supplements 1.5 g per day or placebo for 6 months. The women who were treated with cinnamon supplements had higher homeostasis model assessment (HOMA) and lower quantitative insulin sensitivity check index (QUICK-I), suggesting more insulin sensitivity. This experiment indicated that cinnamon supplementation may be an effective treatment option for some women with PCOS. Swaroop et al. reported that fenugreek seed extract (Furocyst), which is originated from *Trigonella foenumgraecum* L., showed the ability to reduce ovarian volume and the number of ovarian cysts [108]. Two capsules of 500 mg of this compound treated on premenopausal women diagnosed with polycystic ovary syndrome (PCOS) for 90 days enriched in approximately 40% furostanolic saponins and the significant increases in luteinizing hormone (LH) and follicle-stimulating hormone (FSH) levels were observed compared to the baseline values. Furocyst caused significant decrease in both ovarian volume and the number of ovarian cysts, demonstrating the potential of this natural product to enhance reproductive function.

### 2.4. Natural Products with Adverse Effects

On the other hand, several studies demonstrated that some natural products caused adverse effects on fertility (Table 11). Pharm. D. et al. reported that olive fruit extract significantly decreased male fertility parameters [109]. The hydro-alcoholic olive (*Olea europaea*) extract was given orally to SD rats in 50, 150, 450 mg/kg for 48 days. The results showed a significant decrease in the weights of the testicle, seminal vesicle, testosterone hormone, sperms count, and sperm motility. This study concluded that *Olea europaea* extract may have deleterious effects on fertility factors. Khaki et al. reported that permethrin had a negative impact on male fertility [110]. Wistar rats were treated with permethrin (35 mg/kg in 0/5mL dimethyl sulfoxide) via intra-peritoneal injection for 2 months. Permethrin had negative impacts on sperm parameters, the Leydig cells count, and reduced testosterone serum level, decreased sperm motility, inducing damage to spermatozoa. This finding confirmed that permethirn induced adverse effects on male reproductive structures and systems. Adeoye et al. demonstrated testicular toxicity of methanol extract from red cultivar *Allium cepa* Linn., which is a traditional medicine famous for its therapeutic effects on dysentery, chronic bronchitis, and rheumatism [111]. Treated to SD rats, the extract (100, 200, 40, 800, 1200 mg/kg for 14 days) reduced sperm concentration, sperm motility, and sperm mass activity. It was observed that germinal epithelial cells were degenerated and abnormal headless and tailless sperm cells were frequently expressed as abnormal morphology in sperm. These data proposed that administration of *Allium cepa* Linn. at doses more than 100 mg/kg caused spermatotoxic effects and testicular damage, which interferes with normal reproduction in males. Nath reported that *Ricinus communis* L. showed dose dependent loss of sperm motility by influencing the morphological deformation [112]. *Ricinus communis* L. aqueous extract (100, 200, 300 mg/mL) incubated for 30 min with human sperm in vitro condition, was a potent spermicide in comparison to the other three extracts which were petroleumether, ethyl acetate, and acetone. After administration, HOS level and morphological sperm deformity increased, NCD level decreased, and halo formation reduced. This research opened up scope for future exploration of bark of the plant as a source of new male contraceptive. Asuquo reported that *Spondias Mombin* L. possessed significant antifertility effects [113]. *Spondias Mombin* L. ethanol extract (250, 500 mg/kg) was orally administered to wistar strain rats for 8 weeks. The treatment showed decreases in testicular, epididymal weight, and serum level of testosterone, suppressing the process of spermatogenesis, which led to infertility. The results suggested that *Spondias mombin* L. could cause impairment of testicular and epididymal structures which led to significant decrease in spermatogenic activity in seminiferous tubules. Guo et al. investigated the deleterious effect of *Tripterygium wilfordii* Hook. F. (GTW) in both male and female reproductive function [114]. Male and female SD rats, treated with 37.8 or 94.5 mg/kg does for 90 days of *Tripterygium wilfordi* glycoside, revealed increase of immature sperm and decrease of metestrus phase. It was also discovered that levels of ER-α in females decreased and AR in the testis and epididymis decreased as well. These findings indicated that this natural product exerted harmful effects in the testis and the uterus, ultimately leading to adverse effects in both males and females.

## 3. Methods

Literature search was conducted in Pubmed of the National Library of Medicine (www.ncbi.nlm.nih.gov) and Google Scholar (https://scholar.google.com/). Databases were extensively searched on original articles written in English, electronically published until December 2019. The search used the following keywords; ‘male infertility’, ‘male subfertility’, ‘female infertility’, ‘female subfertility’, ‘natural product’, and ‘natural compound’. In vivo, in vitro studies and clinical studies were included while review articles were excluded.

## 4. Discussion

Infertility is the inability to conceive after 12 months or more of regular unprotected sexual intercourse [115]. According to a prevalence survey in 195 countries and regions from 1990 to 2017, infertility is a continuously increasing global burden continuously increasing regardless of gender [116]. Although assisted reproductive technologies are being adopted to increase pregnancy rates, the clear effects and related complications are still in discussion [117,118]. Traditional herbal medicine is used as effective agents to alleviate male and female infertility, and present studies reported high clinical pregnancy rates with improved general status after treatment [119]. Extensive research has been conducted on the profertility effects of natural substances of phytochemicals and mixtures of natural compounds [120,121]. However, investigation is still needed on the physiological actions of natural substances in the human body. Thus, this study explored the profertility effects and related mechanisms of each natural compound and comprehensively interpreted the possibility of using various natural products for infertility.

According to the studies reviewed in this report, natural products that alleviated infertility shared several mechanisms. The efficacy itself was revealed by the regulation of gamete vitality and functional recovery of genital organs. Major mechanisms include recovery of sex hormone imbalance, reduction of oxidative stress, and control of glucose levels. Through a broad understanding of the efficacy, and related mechanisms, the present study extensively identified the potential effects of numerous natural products in alleviating infertility.

### 4.1. Male Reproductive System

The production of healthy gametes is a complex process which requires a series of physiological activities within the reproductive system [122]. The production of sex hormones is a primary step in the formation of gametes. Several studies which are reviewed in this report elucidated the effects of natural products as promoters of sex hormone genesis, which affected normal sperm production. In the case of males, gonadotropic releasing hormones (GnRH) from the hypothalamus stimulates the anterior pituitary gland to produce follicular stimulating hormone (FSH) as well as luteinizing hormone (LH), which subsequently promotes the formation of testosterone [123] (Figure 1). FSH stimulates the proliferation of Sertoli cells, hence is necessary for normal spermatogenesis [124]. LH maintains the level of steroidogenic enzymes while it also mobilizes and transports cholesterol into the inner mitochondrial membrane [125]. The secretion of LH stimulates Leydig cells to transfer cholesterol into the mitochondria by the activation of steroidogenic acute regulatory protein (StAR) and translocator protein (18 kDa; TSPO) [126]. Cholesterol in the mitochondria is then transformed in to pregnelone by CYP11A11, and the subsequent steps stimulated by CYP17A1, 3β-HSD, and 17β-HSD leads to the formation of testosterone [127]. Testosterone is essential in the process of spermatogenesis and inhibition of germ cell apoptosis [128]. It has also been reported that the upregulation of androgens play an important role in the normal function of epididymis in which sperm are matured and mobilized [128,129]. Therefore, the increase of sex hormones and steroidogenic enzymes could demonstrate the proferility properties of natural products associated with steroidogenesis.

### 4.2. Female Reproductive System

The endocrine system also plays an important role in female reproductive function. The hypothalamus pituitary ovary axis (HPO axis) regulates the development of follicles and the ovulation of oocytes (Figure 2). Numerous sex hormones are engaged in the production of oocytes and the fertilization process, hence are used as effective parameters in the examination of various reproductive disorders [130]. FSH and LH are released from the anterior pituitary, and each of the hormones stimulates follicle growth and ovulation. Especially, low levels of LH have been reported to have significant correlation with anovulation, luteal insufficiency, premature oocyte maturation and polycystic ovary syndrome (PCOS), which are prognostic symptoms of infertility [16]. Estrogen and progesterone are positive markers of normal menstruation cycle and the fertilization process. On the other hand, the overproduction of prolactin suppresses the release of GnRH, resulting in reproductive dysfunction. The complex mechanism of female infertility is associated with a number factors including anatomical abnormalities, endocrine problems, and dysfunction of organs [131]. Therefore, the alleviating effects of natural products on female infertility could be explained by the resolution of abnormalities in hormone secretion and reproductive organs.

### 4.3. Oxidative Stress and Fertility

#### 4.3.1. Causes of Oxidative Stress

Oxidative stress is one of the key factors in mediating reproductive function [132]. Environmental contaminants and various man-made chemicals are known to induce oxidative stress, which is addressed as one of the potential risks of infertility [133]. Endocrine disrupting chemicals (EDCs) are well-known factors that stimulate oxidative stress and disrupt hormonal balance, thereby inhibiting normal growth of germ cells [134]. Bisphenol A, formaldehyde, and lead were typically agents that were used to induce reproductive toxicity in models of infertility research [14,25,50]. Especially, bisphenol A is reported to mimic the effects of estrogen in ovarian cells and inducing prostate cancer through cellular proliferation [50,135]. Furthermore, cytotoxic cancer therapies such as cyclophosphamide have detrimental effects upon spermatogenesis [136]. Several natural products in this review were demonstrated as applicable reproductive agents, reducing the notorious effects of EDCs as well as chemotherapy such as mitomycin C, cyclophosphamide, cisplatin, and adriamycin [9,10,14,25,43,46,50,90]. Accordingly, the use of natural materials will be an effective alternative to prevent and treat infertility caused by oxidative stress due to increased exposure to chemicals.

#### 4.3.2. Oxidative Stress and Reproductive Function

Reactive oxidative species (ROS) are one of the critical factors that adversely affect reproductive function in both males and females. Sperm plasma membrane, constituted with unsaturated fatty acids, is especially vulnerable to peroxidative damage. Sperm function deficiency is highly associated with peroxidative damage, which is characterized by loss of motility and failure in sperm–oocyte fusion [137,138] (Figure 3). Free radicals also have the capability to induce DNA damage in sperm either by caspase mediated apoptosis or by the destruction of proteins. Furthermore, oxidative stress has an important role in the female reproductive system, such as endometrial function, oocyte maturation, and folliculogenesis [139]. However, excess ROS from exogenous agents or the lack of antioxidants were found to inhibit ovarian development, interfere with folliculogenesis, and decrease corpus lutea function [140]. These results suggest that it is necessary to maintain adequate levels of oxidative stress in order to maintain reproductive function. Therefore, antioxidants are crucial in the recovery in infertility, and their activities are used as parameters in evaluating the reduction of oxidative stress.

#### 4.3.3. Oxidative Stress Related Enzymes

Superoxide dismutase (SOD) and catalase (CAT) are antioxidative enzymes that convert superoxide anion and peroxide radicals into water and oxygen [141] (Figure 4a). These antioxidants are present in seminal plasma and sperm, and when this natural antioxidant defense is exceeded by ROS, cellular damage occurs. Superoxide radicals are also detoxified through the glutathione pathway in which glutathione peroxide (GPX) is involved in the reduction of hydro-peroxides. Glutathione (GSH) is used as an electron donor and glutathione reductase (GRx) is engaged in this process by converting nicotinamide adenine dinucleotide phosphate (NADPH) into nicotinamide adenine dinucleotide phosphate (NADP). It has been demonstrated that GPx is located within the testis, prostate, seminal vesicles, vas deferens, epididymis, seminal plasma, and spermatozoa [3]. The activation of antioxidant enzymes of CAT, SOD, GPx, and GRx indicates the ability of the natural products to reinforce the defensive metabolism against ROS, and/or protect the reproductive system from toxic assaults. Furthermore, lipid peroxidation induced by ROS induces sperm damage through peroxidation of unsaturated fatty acids in the sperm plasma membrane [142] (Figure 4b). Free radicals react with poly unsaturated fatty acids forming lipid radicals which are then subsequently oxidized into lipid peroxyl radical and lipid peroxide. In order to assess the degree of lipid peroxidation, levels of MDA and TBARS were commonly estimated [143]. Hence, natural products that showed a decrease in the level of MDA could be concluded as potential agents that block lipid peroxidation in gametes, leading to the restoration of normal morphology of sperm. As previous systematic reviews have shown that antioxidants were effective in treating subfertility [14], the effects of natural products found in this study could be expected.

### 4.4. Diabetes and Fertility

Diabetes mellitus (DM) is a prominent endocrine disorder with typical symptoms of hyperglycemia and metabolic disorders. Clinical investigations noted that there is a high prevalence of subfertility in patients with DM [144,145]. Complications of DM such as the imbalance of the hypothalamic–pituitary–gonadal-axis [146], as well as the increase of ROS and DNA fragmentation are prognostic factors of reproductive failure [147]. Furthermore, molecular mechanisms of glucose fluctuation due to DM were discovered to cause detrimental effects on testicular function [148]. High D-glucose concentrations led to alterations in Sertoli cells which are responsible for the production of lactate, a primary source for the maturation of spermatocytes and spermatids [149,150]. However the deprivation of insulin and insufficient glucose consumption were followed by lower lactate production as well as modulations in several glucose transporters [151,152,153]. Thus, existing data showing the regulation of blood glucose levels may open the possibility of natural products to treat DM-associated infertility.

### 4.5. Natural Products and Formulations with Significant Relevance to Male Fertility

In the present report, a total of 81 studies elucidated the potential effects of natural products against male fertility. In total, 75 studies focused on plant derived compounds, five studies on animal derived compounds, and only one study presented a natural substance of fungal origin. Natural products which were mentioned more than once include Curcumin, Echinacoside, and extracts from *Cistanche tubulosa*, *Date palm* pollen, *Diallyl sulfide*, *Lycium barbarum*, Linn., *Lepidium meyenii* Walp., *Morinda oleifera* Lam, and *Tribulus terrestis* Linn [6,9,22,50,51,54,55,63,64,65,104,105]. *Especially, the effects of Tribulus terrestis* Linn. were elicited by in vivo and clinical studies, showing improved sperm motility and spermatozoa performance [9,104,105]. Most experiments examined the structural and functional properties of sperm, such as concentration, motility, and morphology, which are reported as provable reference values by the World Health Organization [154]. Spermatozoa concentration and characteristics of the testis, epididymis, and seminiferous tubules were also examined in order to demonstrate the production capability of particular natural products. These results were found to be related with one or more mechanisms, including hormone balance, antioxidative activities, and lipid metabolism.

Elevation of testosterone and related enzymes such as CYP11A11, CYP17A1, 3β-HSD, and 17β-HSD were detected in a number of studies, further identifying the efficacy of natural products on sperm production. Several natural products including *Cistanche tubulosa* extract, Echinacoside, and *Loranthus micranthus* aqueous methanol extract were reported to increase the levels of these proteins [55,62]. This indicated that these natural products could be utilized to treat patients who have problems with the synthesis of sex hormones. Other natural products which improved the activity of STAR proteins and TSPO were evaluated as potent agents to activate the translocation of cholesterol into the mitochondria. Interestingly, some natural products showed inhibition of estrogen excess, but also showed increase of estradiol levels, which both led to positive results on fertility factors [54]. These results demonstrated that various natural compounds and extracts were capable of regulating steroidogenesis. Moreover, several reports found that the recovery of reproductive ability was related with the downregulation of oxidative stress. The decrease of free radicals like NO and TBRAS, with increases in oxidative enzymes such as SOD, CAT, GSH, GST, and GPx, were observed. This was especially distinguishable in animal models treated with *Acacia hydaspica* ethyl acetate extract, aged garlic extract, Diallyl sulfide, *Loranthus micranthus* aqueous methanol extract, *Carissa opaca* leaves methanolic extract, *Teucrium polium* extract, and *Thymus algeriensis* extract [7,10,32,46,54,62,83]. These results indicated that natural products had the efficacy to reverse infertility induced by oxidative stress. Additionally, some studies examined the models with hyperglycemia or insulin resistance. *Echinacea purpurea* ethanol extract, *Lycium barbarum* polysaccharide, and white tea showed restoration of reproductive function as well as blood glucose levels against STZ-induced diabetic mice [63,84,88]. To add, apoptotic activities by the administration of natural products were observed through the regulation of indicators such as c-caspase3, 8, and bcl-2 [43,54,62]. These studies demonstrated that some natural products had protective effects against cellular damage that were associated with reproductive dysfunction.

Formulas were also identified to have profertility effects in males. Gosha-Jink-Gan is a Kampo formula which is known to manage chemotherapy induced peripheral neuropathy [155,156]. Along with the study our research reviewed, it is newly receiving attention as a potential agent to recover spermatogenesis [93,157]. Saikokaryukatsuboreito is also another Kampo medicine, identified to improve spermatogenesis and testosterone levels [95,158]. Yi Shen Jian Pi is a Chinese herbal decoction used to “fortify the spleen and tonify the kidney.” It has been reported that this formula ameliorated chronic kidney disease, possibly by the modulation of mitochondrial quality control network [159]. Shengjing capsules were also noted to have protective effects against reproductive dysfunction. Other studies have also confirmed its efficacy to activate spermatogonial stem cells by upregulating integrin *α*6/*β*1 expression and improving erectile function through nitric oxide induced relaxation [160,161]. Wuzi Yanzong pill is also a famous Chinese formula that has widely been used to treat kidney essence insufficiency, and studies have demonstrated that this formula alleviated oligoasthenozoospermia by regulating reproductive hormones and TGf-β1/Smads pathway [162]. Subfertile men who were treated with this formula showed improved semen qualities compared to the control group, stating the necessity for further studies that could verify the efficacy of Wuzi Yanzong pill on a broader scale [163]. Since the efficacy of these formulation is previously proven by numerous case reports, practical utilization of these products in the near future seems to be promising.

### 4.6. Natural Products with Significant Relevance to Female Fertility

Reports on natural products affecting female fertility were relatively scarce, with only 17 studies dealing with plant derived substances, and one study dealing with an animal derived substance. The only natural compound that was mentioned more than once was cinnamon, showing positive effects in insulin resistance [99,107]. *Apis mellifera* was the compound that was indicated in both male and female studies, which was mentioned in four different reports [2,21,89,90,102]. This compound was formerly demonstrated to show antioxidant, anti-hyperglycemic, and antidiabetic activities [164]. However, the efficacy should be further studied in order to demonstrate the similarities between the reports, since the forms of the extract were all different. The effects of natural substances on female infertility have been revealed through experimental models with diverse underlying conditions. There were natural products that restored reproductive function by resolving structural diseases such as endometrial hyperplasia, endometritis, and PCOS, which is one of the most well-known causes of ovulatory infertility [165]. All of the three clinical studies were conducted on PCOS patients, which supports the possibility of the natural products as protective agents against PCOS related infertility [106,107,108]. On the other hand, other materials were discovered to normalize ovarian function and menstrual cycle, and increase implantation success rate [12,98]. The natural products that were reviewed in this study showed elevation in FSH and LH, as well as significant increases in estrogen and progesterone which supported the overall promotion of reproductive activity [20,106,130,166]. In addition, increased integrin and LIF were observed, and these mechanisms were found to be effective in alleviating existing diseases, recovering menstrual cycles, ovarian function, and creating optimal conditions for implantation. As in male infertility, a decrease in free radicals and an increase in antioxidant enzymes were observed in reports with natural products such as barley and royal jelly [98,102]. In addition, there have been experiments that showed that natural substances that helped control cholesterol and increase insulin sensitivity, which include cinnamon and *Ficus asperifolia* aqueous extract [13,99]. As a result, it was confirmed that the reduction of oxidative stress and control of lipid components were common factors that were regulated by natural products, improving both male and female fertility.

### 4.7. Natural Products with Adverse Effects

On the other hand, several natural products were noted to exert reproductive toxicity. Administration of high doses of extracts resulted in adverse effects in fertility. The dose of *Ricinus communis* L. aqueous extract was higher than 100 mg/mL, and the dose of *Spondias mombin* L. ethanol extract and Red cultivar *Allium cepa* methanol extract were higher than 100 mg/kg [111,112,113]. These studies used a reasonably high enough to cause toxicity, and further research at lower levels is necessary. However, *Olea europaea* hydro-alcoholic extract and Permethrin showed a decrease in reproductive function even in doses as low as 50 and 35 mg/kg, respectively [109,110]. *Tripterygium glycoside* extract was mentioned to have adverse effects in both male and female rats, inhibiting the activity of hormone receptors critical in the process of fertilization [114]. According to previous reviews, several natural products and their substances showed antifertility effects, and were even mentioned as new alternatives of male contraceptives [167,168,169]. It is necessary to identify certain natural products that show contradictory effects upon reproductive function, and to investigate the changes in effects according to the method of drug production and administration.

### 4.8. Insufficient Data of Existing Research

Some reports lacked prior toxicology experiments in order to examine stability of the natural products [1,2,9,48,63]. This was frequently seen in studies that were based on formulas, since these products were previously reported to have been used in human patients [8,30,96]. Moreover, in several studies, the control and experimental groups were not properly set [34]. These studies only identified the alleviating effects of infertility compared to control groups which were previously induced with various toxic substances. In this view, studies are required to secure the basis for development of natural products as profertility agents. In addition, the study on the effect of inositol only revealed histological results of sperm, which is insufficient to verify the effectiveness of the natural product [103]. Meanwhile, some experiments were conducted on normal cells or animals [17,40,42,45,76,104,170]. These results are insufficient to support whether the natural products have the capacity to protect genital damage or reverse reproductive disorders. In order to fully demonstrate the efficacy of the natural products, it is necessary to progress a properly designed experiment that is targeted on models that induced reproductive toxicity in advance. Moreover, several studies were progressed with high doses, especially maca capsule and extract were treated using more than 500 mg/kg [64,65]. Two of three studies that reported the efficacy of *Moringa oleifera* used more than 100 mg/kg [66,67]. Additionally, *Nigella sativa* seed powder was treated using 300 mg/kg [69], while Diallyl sulfide and *Matricaria chamomilla* hydroethanolic extract were treated using over 200 mg/kg [25]. High concentrations have the risk of inducing toxicity or causing adverse effects. Hence the validity of the results should be reconsidered, and further research of the effects associated with reducing oxidative stress and improving sexual activity is needed.

### 4.9. Limitations of This Review and Future Prospects

One of the limitations of our study is the absence of clinical trials. Although there have been experiments with human-derived cells, actual clinical trials have not been reviewed. Two clinical trials with *Alpinia officinarum* Hance rhizome extract and *Withania somnifera* were estimated to show enhancement in spermatogram factors upon males suffering from idiopathic infertility [170,171]. However, these studies lacked sufficient numbers of patients and the set of control groups was unclear, which did not meet the inclusion criteria of our study. This shows that there is a lack of in-depth clinical studies on drugs that are of natural origin, and further clinical research in this field is necessary. Furthermore, our study attempted to review both studies on male and female infertility, but unfortunately there have been few studies on female infertility. In addition, the mechanisms of natural products to alleviate infertility were diverse but in-depth investigations were only restricted to hormones and oxidative stress. For instance, *Cyperus rotundus* water extract and *Perilla frutescens* treatment showed an increase in LIF levels, which is a cytokine known to regulate embryo implantation [12,15,172]. A more in-depth investigation in the various mechanisms in female infertility should be assessed in further research. Another limitation of our study is that it did not distinguish the mode of drug administration, which might affect the efficacy of the drug. Oral administration was the majority, however, intradermal injection was also used [90]. In some cases, the products were utilized as a powder form, mixed in daily diets for animals [1,17]. In order to expand the comprehensive review of natural substances into studies on actual clinical applications, it is expected that analytical studies on the mechanism and range of use of drugs will be required.

This review broadened the interpretation of natural products as potential drugs to alleviate infertility and reproductive insufficiency. However, additional studies are necessary in the determination of safety and efficacy based on pharmacokinetic research. Standardization of dosage and extraction methods are also needed for the utilization in clinical fields. Additionally, more rigorous examination on the mechanism in fertility is required. In the case of ROS, current studies identified the bilateral aspects of ROS in spermatogenesis working as activators as well as inhibitors [173]. Novel mechanisms such as the proliferation of SSC by 5H-purin-6-amine, PI3K/AKR, SFK signaling pathways, and the lack of Nrf2 genes were reported to probably affect arsenic-induced reproductive toxicity [41,59]. Moreover, the fluctuation of hormones such as FSH, are caused by more than one physiological process, therefore the hormonal changes by certain natural products should be interpreted with a wider variety of possibilities [174,175]. Since reproductive function is a complex process and the related effects of natural substances are also diverse, further in-depth exploration is expected.

In conclusion, our study reviewed various natural products that showed profertility effects through gamete performance and the function of reproductive organs. These results were identified by the regulation of sex hormones, oxidative stress, lipid metabolism, as well as the apoptosis pathway. Natural products from plant, animal, and fungal origin, as well as formulations were widely covered. Although several limitations exist, the anti-infertility effects of various natural products and the detailed mechanisms have opened up new possibilities in the treatment of infertility.

## 5. Conclusions

In conclusion, this study aimed to investigate natural products that showed effective improvement on infertility and subfertility in men and women. Natural compounds, extracts, and various formulations were discovered to show efficacy in the production of gonadotropic hormones and the activation of antioxidative processes including lipid peroxidation and glutathione synthesis. Several natural products also showed efficacy in the regulation of glucose and the apoptotic pathways. This study deals with natural substances, extracts, and prescriptions of various origins, and attempted to establish a balanced view by examining drugs that showed opposite (contraceptive) effects. However, there are still limitations in that research on infertility lacks clinical studies and the dosage and utilization methods of formulations were indistinguishable. Future studies are expected to clarify the mechanisms of the profertility effects and to refine pharmacological effects of natural products for clinical use.

## Figures and Tables

**Figure 1 antioxidants-09-00957-f001:**
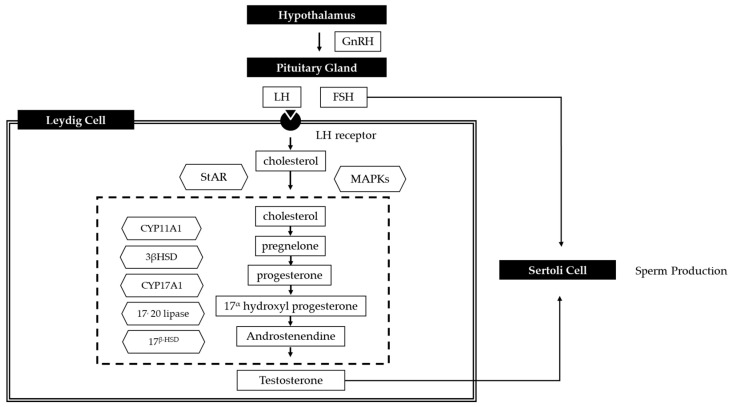
Male sex hormone formation. GnRH is secreted from the hypothalamus, subsequently LH stimulating the Leydig cell to produce testosterone and FSH activating the Sertoli cells to induce sperm production.

**Figure 2 antioxidants-09-00957-f002:**
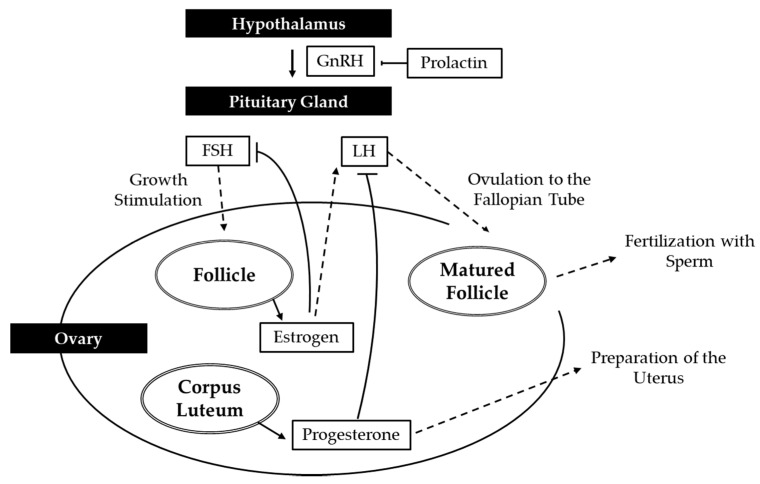
Female sex hormone formation. GnRH secreted from the hypothalamus induces the production of FSH from the pituitary gland to stimulate follicle growth. Estrogen induces the production of LH which ovulates the mature follicle, while progesterone made from the corpus luteum inhibits the activation of LH.

**Figure 3 antioxidants-09-00957-f003:**
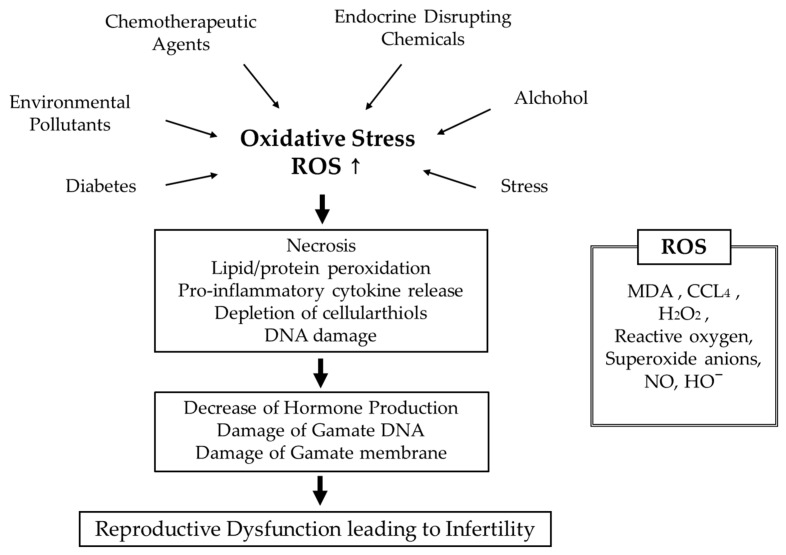
Reactive oxidative species (ROS) and infertility. ROS are induced from a variety of causes and consequently destroy the structure of lipid matrix in the membranes of spermatozoa and are associated with loss of motility and impairment of spermatogenesis.

**Figure 4 antioxidants-09-00957-f004:**
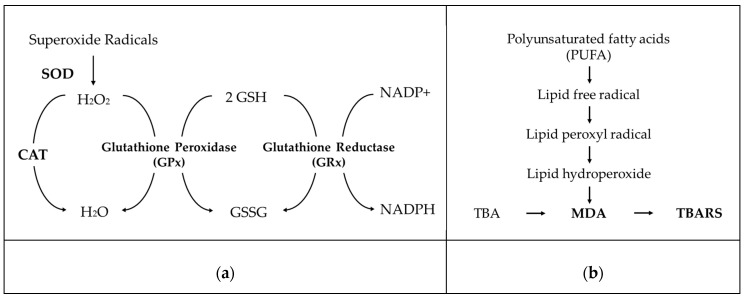
Glutathione oxidation pathway and lipid peroxidation pathway. (**a**) The glutathione pathway detoxifies superoxide radicals through the reduction of hydro-peroxides by glutathione peroxide (GPX). (**b**) Lipid peroxidation pathway is induced by ROS and MDA and TBRAS which are the products of the pathway used to assess the degree of the oxidation process.

**Table 1 antioxidants-09-00957-t001:** Plant derived natural products and male infertility (in vitro studies).

Classification	Compound/Extract	System	Source	Cell Line/Animal Model	Dose; Duration	Efficacy	Mechanism	Reference
Plant	Date palm pollen extract	In vitro	*Phoenix dactylifera* Linn.	Sertoli cells, spermatogonial stem cells from mice	0.06, 0.25, 0.62 mg/mL; 14 days	Increase of proliferation of spermatogonia		[40]
Plant	5H-purin-6-amine, *Sedum sarmentosum* extract	In vitro	*Sedum sarmentosum*	Spermatogonial stem cells C57BL. 6-TG-EGFP	0.01, 0.1, 1, 10 mg/mL; 1 week	Increase of self-renewal in SSCs	↑ PLZF, GFRα1, VASA, Lhx1 ↓ Pgk2	[41]
Plant	Licorice extract	In vitro	*Glycyrrhiza uralensis* Fisch.	Testis tissue from C57BL/6N mice	0.2, 2, 20 μmol/L; 72 h	Increase of proliferation of spermatogonia	↑ PCNA, SCP3, Spo11	[42]
Plant	*Lycium barbarum* polysaccharide	In vitro	*Lycium barbarum* Linn.	Leydig MLTC-1	50 μg/mL; 48 h	Increase of cell viability	↑ Testosterone, ↓ p-PERK/PERK, p-elF2α/elF2α, ATF4/β-actin, apoptosis rate, LC3II/I, Atg5/β-actin	[43]
Plant	*Morindae* radix aqueous extract	In vitro	*Morinda officinalis*	TM3 cells, mouse Leydig cells	10, 50, 100, 250 mg/mL; 24 h	Increase of testosterone production. Decrease of H_2_O_2_ induced cytotoxicity, and lipid peroxidation	↑ SOD, CAT ↓ MDA	[44]
Plant	*Taraxacum officinale* aqueous extract	In vitro	*Taraxacum officinale*	TM3, ATCCNoCRL	1, 10, 25, 50 mg/mL; 12, 48 h	Increase of the levels of steroidogenic enzymes	↑ STAR, CYP11A1, CYP17A1	[11]
Plant	*Typha capensis* rhizome extract F1 fraction	In vitro	*Typha capensis* (*Rohrb.*) *N.E.Br.*	TM3-Leydig cells	10, 100 μg/mL; 96 h		↑ Testosterone	[45]

PCNA, proliferating cell nuclear antigen; PERK, protein kinase-like endoplasmic reticulum kinase; p-PERK, phospho-PERK; elF2α, eukaryotic initiation factor 2; p-elF2α, phospho-elF2α; ATF4, activating transcription factor 4; SCP3, synaptonemal complex protein 3; SOD, superoxide dismutase; CAT, catalase; MDA, malondialdehyde; StAR, steroidogenic acute regulatory protein; CYP17A1, cytochrome P450 17A1; CYP11A1, cytochrome P450 11A1.

**Table 2 antioxidants-09-00957-t002:** Plant derived natural products and male infertility (in vivo studies).

Classification	Compound/Extract	System	Source	Cell Line/Animal Model	Dose; Duration	Efficacy	Mechanism	Reference
Plant	*Acacia hydaspica* ethyl acetate extract	In vivo	*Acacia hydaspica* R. Parker	SD rats	400 mg/kg; 21 days	Increase of seminiferous tubule diameter, area, epithelial height Decrease of width of tubular lumen, interstitial space, DNA damage	↑ Testosterone, LH, FSH, SOD, POD, CAT, QR, GSH, GR, GST, GPx, γ-GT ↓ H_2_O_2_, NO, MDA	[46]
Plant	*Achillea millefolium* inflorescences ethanol extract	In vivo	*Achillea millefolium* Linn.	Wistar mice	120 mg/kg; 48 days	Increase of sperm motility, capsule thickness, epithelial thickness, tubule differentiation index	↑ SOD, LH ↓ LDH, NO, MDA	[47]
Plant	Aged garlic extract	In vivo	*Allium sativum* for. *pekinense MAKINO*	Wistar mice	250 mg/kg; 14 days	Increase of testis weight, sperm count, motility, recovery of seminiferous tubules Decrease of the death of sperm, sperm abnormality	↑ Testosterone, GSH, GSH-Px, CAT, SOD ↓ MDA	[10]
Plant	*Angelica keiskei* powder	In vivo	*Angelica keiskei* Koidz.	Self-breeding CD-1 mice	57.5 mg/kg; 7 days	Increase of density of sperm, motility, motile sperm density, progressive sperm velocity, progressive sperm density Decrease of abnormal seminiferous tubules, DNA fragmentation	↑ GSS, HO-1, Hspa11, Hspa2 Hsf1, Hsf2	[1]
Plant	Banaba leaf and ginseng extract	In vivo	*Lagerstroemia speciosa* (L.) Pers.	Swiss mice	150 mg/kg; 30 days	Increase of testis weight, epididymis weight, seminal vesicle weight, sperm density, sperm viability, progressive sperm motility Decrease of nonprogressive sperm motility, abnormal sperm morphology (head, tail, twisted body)	↑ Testis glycogen, testis protein, testis fructose, seminal vesicle fructose, epididymal fructose, testis protein, seminal vesicle protein, epididymal protein ↓ Testis cholesterol	[37]
Plant	Bajijiasu	In vivo	*Morinda officinalis* F.C.	(1) Normal Kunming mice kidney-yang-deficient Kunming mouse	(1) 20, 40, 80, 160, 320 mg/kg; 30 days (2) 80, 160, 320 mg/kg; 18 days	Increase of the sexual behavior Decrease of DNA damage of sperm by H_2_O_2_	↑ Testosterone, ↓ Cortisol, SOD, GPx, CAT, MAD	[6]
Plant	*Balanites aegyptiaca* sapogenin extract	In vivo	*Balanites aegyptiaca*	Albino rats	25, 50, 100 mg/kg; 70 days	Increase of semen quality	↑ LH, Estradiol, Testosterone, Glucose ↓ FSH, Cholesterol, sAST, urea, creatinine	[48]
Plant	Bee bread	In vivo	Bee pollen	SD rats	0.5 g/kg; 28 days	Increase in the weight of the prostate gland in adult rats	↑ Arginine, L-carnitine, glutathione	[49]
Plant	*Cistanche tubulosa* extract	In vivo	*Cistanche tubulosa* (Schrenk) R. wight	SD rats	200 mg/kg; 42 days	Increase of daily sperm production Decrease of abnormal morphology (isolated head, head without curvature), immobile sperm	↑ LDH-x activity, testosterone, 3β-HSD, 17β-HSD, CYP17A1, CYP11A1, StAR	[50]
Plant	*Crocus sativus* L. aqueous extract	In vivo	*Crocus sativus* Linn.	Wistar rats	100 mg/kg; one time/2 days; 16 days	Increase of mean sperm number	↑ Testosterone ↓LPO	[27]
Plant	Curcumin	In vivo	Turmeric	SD rats	100 mg/kg; 3 days	Decrease of the toxic effects of CdCl_2_	↑ GSH, CAT, GPx, SOD ↓ TBARS	[51]
Plant	Curcumin extract	In vivo	*Curcuma longa* Linn.	Bovine semen breeding bulls	5, 10, 50, 100 µM/L; 24 h	Increase of spermatozoa activity and protection	↓ Nitroblue-tetrazolium	[52]
Plant	*Cymbopogon citrates* aqueous extract	In vivo	*Cymbopogon citratus*	SD rats	100 mg/kg; 30 days	Decrease of H_2_O_2_ induced reproductive system injury	↑ GSH ↓ MAD	[4]
Plant	Date palm pollen extract	In vitro	*Phoenix dactylifera* Linn.	Sertoli cells, spermatogonial stem cells from mice	0.06, 0.25, 0.62 mg/mL; 14 days	Increase of number of spermatogonial colony		[40]
Plant	Date Palm Pollen extract	In vivo	*Phoenix dactylifera* Linn.	Wistar mice	150 mg/kg; 56 days	Decrease of testicular dysfunction	↑ LH, FSH, Testosterone, 3β-HSD, 17β-HSD Estradiol	[53]
Plant	Diallyl sulfide (DAS)	In vivo	*Allium sativum* Linn.	Swiss albino rats	200 mg/kg; 49 days	Increase of sperm counts, weight of testis, epididymis, spermatogenesis	↑ Testosterone, estradiol, CYP19, SOD, GSH ↓ MDA, NO	[54]
Plant	*E. amoenum* distillate	In vivo	*Echium amoenum*	Mus musculus mice	150 ± 2.5, 75 ± 1.25 mL/kg; 3 weeks	Improvement of hormonal and sperm parameters	↑ FSH, LH, Testosterone, Leydig cells	[34]
Plant	*Echinacea purpurea* extract	In vivo	*Echinacea purpurea* Linn.	Albino rats	63 mg/kg; 4 weeks	Improvement of sperm parameters in the oxidative stress	↑SOD, GST calcium ion ↓MDA, NO	[26]
Plant	Echinacoside	In vivo	*Cistanche tubulosa Hook f. II.*	Kunming mice	5, 20, 80 mg/kg; 14 days	Increase of epididymal sperm count, sperm motility	↑ LH, CYP11A1, CYP17A1, HSD3β1/2, HSD17β, StAR, LHβ, LHr, Gnrh 1, Gnrhr ↓ Hypothalamic AR in nuclei	[55]
Plant	Echinacoside	In vivo	*Cistanche tubulosa* (Schrenk) R. wight	SD rats	6 mg/kg; 42 days	Increase of sperm number in testis, daily sperm production Decrease of abnormal morphology (isolated head, head without curvature), immobile sperm	↑ LDH-x activity, testosterone, LH, FSH, 3β-HSD, 17β-HSD, CYP17A1, CYP11A1, StAR	[50]
Plant	Ethyl pyruvate	In vivo	NO	NMRI mice	40 mg/kg; 35 days	Decrease of destructive effects of PHZ on sperm parameters, testosterone level, and lipid peroxidation	↓ MDA	[29]
Plant	*Eurycoma longifolia* extract	In vivo	*Eurycoma longifolia*	SD rats	8 mg/kg; 14 days	Increase of testicular function, spermatogenesis, sperm counts, and motility Decrease of the effects of an excessive estrogen state	↓ Estrogen	[56]
Plant	*Ginkgo biloba* extract	In vivo	*Ginkgo* *biloba* Linn.	Wistar mice	50 mg/kg; one time	Increase of seminiferous tubular diameter, primary spermatocyte number, round spermatid number, Leydig cell number	↑ Testosterone, FSH ↓ Mitochondrial NAD, plasma TNF-α, plasma IL-1β	[57]
Plant	Ginger	In vivo	*Zingiber officinale*	Wistar rats	50, 100 mg/kg; 20 days	Increase of sperm healthy parameters	↑ TAC, LH, FSH ↓ MDA	[58]
Plant	Grape seed proanthocyanidin extract	In vivo	*Vitis vinifera*	Kunming mice	100, 200, 400 mg/kg; 5 weeks	Decrease of oxidative stress damage in mice testis	↑ Nrf2, GST, HO1, NQO1, SOD, T-AOC ↓ MDA, 8-OHdG	[59]
Plant	*Ionidium suffruticosum* methanol extract	In vivo	*Ionidium suffruticosum* (L.) Ging (Violaceae)	Albino rats	250 mg/kg; 28 days	Increase of sperm count, cauda epididymis sperm motility, body weight, germinal epithelial cell mass, testis weight, sperm vitality Decrease of epidermal sperm agglutination, sperm morphology abnormality (detached tail, fusion of sperm, broken middle piece, detached tail, coiling of flagellum)	↑ CAT, SOD ↓ MDA	[60]
Plant	*Jurenia dolomiaea* methanol extract	In vivo	*Jurenia dolomiaea* Boiss.	SD mice	200, 400 mg/kg; 60 days	Increase of thickness in germinal layers	↑ SOD, CAT, POD, testosterone, GSH, GST, GPx, GR, ↓ H_2_O_2_, TBARS, Nitrate	[61]
plant	KH-465	In vivo	*Epimedium koreanum* Nakai, *Angelica gigas* Nakai	SD rats	200, 400 mg/kg; 4 weeks	Increase of the sperm count and motility	↑ LH, SOD ↓ 8-OHdG	[5]
Plant	*Loranthus micranthus* aqueous methanol extract	In vivo	*Loranthus micranthus* Linn.	Wistar rats	100, 200 mg/kg; 14 days	Increase of testis weight, sperm motility, sperm viability, TSN, seminiferous tubule diameter, Leydig cells count Decrease of sperm abnormality	↑ Testosterone, FSH, LH, 3β-HSD, 17β-HSD, SOD, CAT, GSH, GSH-Px, GST, Bcl-2 ↓ MDA, LPO	[62]
Plant	*Lycium barbarum* polysaccharide	In vivo	*Lycium barbarum*, Linn.	ICR mice	20, 40 mg/kg; 62 days	Increase of testis weight, epididymis weight, testis organ coefficient, epididymis coefficient, sperm count, sperm viability, mating rate, fertility rate, recovery of spermatogonia, recovery of Sertoli cells	↑ Testosterone, FSH, LH	[63]
Plant	Maca capsules	In vivo	*Lepidium meyenii* Walp.	Empire Breeders mice	500, 1000 mg/kg; 28 days	Increase of sperm count, sperm motility, seminiferous tubule width, germinal cell layer thickness	↑ Testosterone, GSH, CAT, SOD ↓ MDA	[64]
Plant	Maca extract	In vivo	*Lepidium meyenii* Walp.	BALB/c mice	666 mg/kg; 35 days	Increase of sperm motility, sperm count		[65]
Plant	*Matricaria chamomilla* hydroethanolic extract	In vivo	*Matricaria chamomilla*	Wistar rats	200, 500 mg/kg; 30 days	Increase of testosterone, LH Enhancement of sperm counts, motility, viability	↑ PI3k, Akt	[25]
Plant	*Mentha spicata* aqueous extract	In vivo	*Mentha spicata*	Albino mice	40, 100, 400 mg/kg; 1 week	Decrease of ifosfamide induced chromosomal aberration in bone marrow cells of male albino mice	↓ Ifosfamide	[36]
Plant	*Carissa opaca* leaves (MLC) methanolic extract	In vivo	*Carissa opaca* leaves	SD rats	50, 100, 200 mg/kg; 8 weeks	Protective effect against CCl4-induced antioxidant and hormonal dysfunction	↑ CAT, POD, SOD, GST, GPx, GR, GSH, QR ↓ Triglycerides, cholesterol, HDL, LDL, TBARS, H_2_O_2_	[7]
Plant	*Moringa oleifera* Seed Aqueous Extract	In vivo	*Moringa oleifera*	Wistar strain albino rats	(1) 1000, 2000, 5000 mg/kg; 7 days (2) 100, 200, 500 mg/kg; 21 days	Enhancement of sexual behavior		[66]
Plant	*Moringa oleifera* Lam. leaf powder	In vivo	*Moringa oleifera* Lam.	New Zealand White rabbits	5, 10, 15 g/kg; 12 weeks	Increase of semen volume, sperm count, motility Decrease in abnormal morphology of sperm	↑ FSH, LH	[17]
Plant	*Moringa oleifera* Lam. extract	In vivo	*Moringa oleifera* Lam.	SD rats	400, 800 mg/kg; 2 weeks	Increase of germinal cell layer thickness, diameter of seminiferous tubules, testis weight index, testicular weight Decrease of perivascular fibrosis	↑ SOD ↓ MDA, HSP70,	[67]
Plant	(1) *Marjoram* (2) Sage	(1) In vivo (2) In vivo	(1) *O. Marjoram* (2) *S. officinalis*	Wister albino rats	(1) 0.16 mL/kg; 12 weeks (2) 0.05 mL/kg; 12 weeks	Increase of androgen, sperm count, and improvement of testicular structure Decrease of testicular lipid accumulation	↑ Testosterone, DHEA, T/E2 ↓ Leptin, PRL, E2	[23]
Plant	Naringenin	In vivo	Citrus species	SD rats	40, 80 mg/kg; 10 weeks	Increase of progressive motility, seminiferous tubule lumen volume Decrease of seminiferous epithelium volume		[68]
Plant	*Nigella sativa* Seed Powder	In vivo	*Nigella sativa*	Albino rats	300 mg/kg; 45 days	Increase of testosterone levels	↑ LH	[69]
Plant	*Pedalium murex* methanol fruit fraction	In vivo	*Pedalium murex* Linn.	Albino rats	50, 10 mg/kg; 60 days	Increase of fertility, sperm motility, sperm density, spermatogenesis, germinal cell count, interstitial cell count, spermatid, preleptotene spermatocyte, fibroblast, mature Leydig cell	↑ LH, FSH, testosterone, cholesterol, glycogen, sialic acid	[70]
Plant	*Petasites japonicus* MeOH extract	In vivo	*Petasites japonicus*	SSC/ C57BL/6 mice	0.1, 1, 10 μg/mL; 7 days	Increase of spermatogonial stem cells	↑ LHX1, GFRα1	[71]
Plant	*Phoenix dactylifera* date palm pollen extract	In vivo	*Phoenix dactylifera*	SD rats	30, 40, 120, 240 mg/kg; 35 days	Increase of sperm count, motility, morphology, DNA quality	↑ Estrogen, testosterone	[72]
Plant	*Phyllanthus* *emblica* L. extract	In vivo	*Phyllanthus* *emblica* Linn.	SD rats	50 mg/kg; 42 days	Increase in sperm concentration, testicular size Decrease in sperm head abnormality, acrosome-reacted sperm	↑ Testosterone, StAR ↓ Corticosterone, MDA,	[73]
Plant	*Pilea microphylla* extract	In vivo	*Pilea microphylla* (L.) Liebm.	Wistar mice	50 mg/kg; 10 weeks	Increase of left epididymal sperm count, motility, vitality, and morphology		[74]
Plant	*Pistia stratiotes* Linn. ethanol extract	In vivo	*Pistia stratiotes* Linn.	Wistar strain albino mice	100 mg/kg; 14 days	Increase of sperm motility Decrease of sperm abnormality		[75]
Plant	*Rosa damascene* aqueous extract	In vivo	*Rosa damascena* Linn.	NMRI mice	10, 20, 40 mg/kg; 40 days	Increase of sperm number, sperm motility, sperm viability, rate of normal sperm, testis weight, testis length, testis width, number of Leydig cells	↑ Testosterone	[33]
Plant	Safed musli extract	In vivo	*Chlorophytum borivilianum* Santapau and Fernandes	Wistar mice	125, 250 mg/kg; 52 days	Increase of sperm count, mount latency		[76]
plant	Sanrego aqueous extract	In vivo	*Lunasia amara*	SD rats	0, 30, 60, 90 mg/kg; 42 days	Increase of sperm number, sperm motility		[77]
Plant	Shilajit water extract	In vivo	*Shilajit*	Swiss albino mice	50, 100, 200 mg/kg; 35 days	Increase of weight of testis, epididymis, seminal vesicle, sperm production, sperm motility, sperm concentration, sialic acid concentration in epididymis, fructose concentration in seminal vesicle, libido, male fertility index Decrease of affected seminiferous tubules	↑ Testosterone, 3β-HSD, 17β-HSD	[78]
Plant	Silymarin	In vivo	*Silybummarianum* seed	Wistar rats	50, 100, 150 mg/kg; 28 days	Increase of spermatid and spermatozoid cells	↑ FSH, GnRH, LH, testosterone, GPX, SOD, NE, serotonin, dopamine	[79]
Plant	Suruhan ethanol extract	In vivo	*Peperomia pellucid* L. Kunth	Albino mice	56, 112, 168 mg/kg; 35 days	Increase of sperm counts, viability, motility and morphology recovery Decrease of blood glucose levels	↓ Glucose	[80]
Plant	*Tetracarpidium conophorum* leaf extract	In vivo	*Tetracarpidium conophorum* (Mull. Arg.) Hutch and Dalziel	Wistar mice	500, 1000 mg/kg; 21 days	Increase of testis weight, sperm concentration, sperm viability, sperm motility, normal chromatin integrity Decrease of total sperm abnormality (tailless head, headless tail, bent tail), abnormal chromatin integrity	↑ G6PD activity, 3β-HSD, 17β-HSD, testicular glycogen, testicular Zn, Se content, epididymal Zn, Se content, FSH, LH, testosterone ↓ Testicular cholesterol	[81]
Plant	Walnut leaf extract	In vivo	*Tetracarpidium conophorum* (Müll.Arg.) Hutch. and Dalziel	Wistar albino mice	500, 1000 mg/kg; 21 days	Increase of testis weight, epididymis weight, sperm count, sperm motility, curvilinear velocity, epididymal sperm viability Decrease of sperm abnormality (tailless head, headless tail, bent tail, coiled tail)	↑ Testosterone, FSH, LH, 3β-HSD, 17β-HSD, G-6PDH, LDH, testicular Zn and Se content, epididymal Se content, testicular glycogen ↓ testicular cholesterol	[82]
Plant	*Teucrium polium* extract	In vivo	*Teucrium polium* Lam.	Wistar rats	1 mL/kg; 10 weeks	Increase of sperm motility, sperm count Decrease of sperm abnormality	↑ Testosterone, FSH, LH, GPx, CAT, SOD ↓ TBARS	[32]
Plant	*Thymus algeriensis* extract	In vivo	*Thymus algeriensis* Lam. (Boiss. et Reut.)	Wistar rats	150 mg/kg; 2 weeks	Increase of sperm count, sperm viability, sperm motility, normal sperm morphology Decrease of sperm abnormality, DNA fragmentation	↑ CAT, SOD, GPx, GSH, GST ↓ LPO	[83]
Plant	*Tribulus terrestris* dry extract	In vivo	*Tribulus terrestris* Linn.	Swiss albino mice	11 mg/kg; 14 days	Increase of sperm motility	↑ RS, TBRAS, SOD, CAT, GPx, GST, testosterone, 17*β*-HSD	[9]
Plant	WTEA	In vivo	White tea	Wistar rats	1 g/100 mL; 2 months	Increase of sperm concentration, sperm viability, sperm motility Decrease of abnormal sperm morphology	↑ Insulin sensitivity, glucose tolerance, FRAP ↓ Insulin resistance, testis carbonyl content, testicular antioxidant potential, testicular OS, TBARS	[84]
Plant	Xanthoangelol	In vivo	*Angelica keiskei* Koidz.	Self-breeding CD-1 mice	3 mg/kg; 7 days	Increase of sperm motility, progressive sperm density, progressive sperm velocity Decrease of abnormal seminiferous tubules	↑ GSS	[1]
Plant	*Xanthosoma sagittifolium*	In vivo	*Xanthosoma sagittifolium* Linn.	Wistar mice	25, 50, 75, 100%, 100 mg/kg; 14 days	Increase of sperm count, sperm motility, sperm livability Decrease of luminal diameter		[85]
Plant	*Tetracarpidium conophorum* leaf extraxct	In vivo	*Tetracarpidium conophorum*	Wistar mice	50, 500, 1000 mg/kg; 21 days	Decrease of oxidative reproductive toxicity	↑ MDA, GSH	[86]
Plant	*Zingiber officinale* aqueous extract	In vivo	*Zingiber officinale*	Broiler breeder	5, 10 %; 64 weeks	Increase of spermatogenesis	↑ FSH, Testosterone, LH ↓ MDA, TAC	[87]

Cyp19, P450 enzyme aromatase; SOD, superoxide dismutase; GSH, glutathione; MDA, malondialdehyde; NO, nitric oxide, GSS, glutathione synthase, GST, glutathione-S-transferase; GPx, glutathione peroxidase; GSH-Px, glutathione peroxidase; Hsp70, heat shock protein 70, CAT, catalase; RAGE, receptor for advanced glycation end products; NF-*κB*, nuclear factor kappa-light-chain-enhancer of activated B cells; H_2_O_2_, hydrogen peroxide; TNF-α, tumor necrosis factor α; IL-6, interleukin 6; IL-1β, interleukin 1 beta; TBRAS, thiobarbituric acid reactive substances; RS, reactive species; POD, peroxidase; QR, quinone reductase; GR, glutathione reductase; γ-TG, γ-glutamyl transpeptidase; StAR, steroidogenic acute regulatory; G6PD, glucose-6-phosphate dehydrogenase; G6PDH, glucose 6-phosphate dehydrogenase; CYP17A1, cytochrome P450 17A1; CYP11A1, cytochrome P450 11A1; LPO, lipid peroxidation; NAD, nicotinamide adenine dinucleotide; Gnrhr, gonadotropin-releasing hormone receptor; LH, luteinizing hormone; LHβ, luteinizing hormone β subunit; LHγ, luteinizing hormone γ subunit; FSH, follicle-stimulating hormone; Nrf2, nuclear factor erythroid 2-related factor 2; HO-1, heme oxygenase-1; 3β-HSD, 3β-hydroxysteroid dehydrogenase; 17β-HSD, 17β-hydroxysteroid dehydrogenase; HSD17β3, hydroxysteroid dehydrogenase 17β3; HSF1, heat shock factor 1; HSF2, heat shock factor 2; AR, androgen receptor; TLR2, tool like receptor 2; TLR4, tool like receptor; MDC, macrophage-derived chemokine; LC3, microtubule-associated protein 1A/1B-light chain 3; LDH, lactate dehydrogenase; Bcl-2, B-cell lymphoma 2; Bax, BCL2-associated X Protein; BrdU, 5-bromo-2-deoxyuridine; FRAP, ferric reducing ability of plasma; LHX1, LIM homeobox 1; GFRα1, glial cell line-derived neurotrophic factor family receptor α1; Atg, autophagy-related; Zn, zinc; Se, selenium.

**Table 3 antioxidants-09-00957-t003:** Plant derived natural products and male infertility (in vitro and in vivo studies).

Classification	Compound/Extract	System	Source	Cell Line/Animal Model	Dose; Duration	Efficacy	Mechanism	Reference
Plant	*Echinacea purpurea* ethanol extract (encapsulated chitosan/silica nanoparticle)	In vitro and in vivo studies	*Echinacea purpurea* Linn.	(1) LC540 (2) SD rats	(1) 25 µg/mL; 24 h (2) 279, 465 mg/kg; 7 weeks	(2) Increase of seminiferous tubules diameter, germinal cell layer thickness, area of seminiferous tubules, area of seminiferous lumen, sperm motility, sperm DNA integrity Decrease of sperm abnormality	(1) ↓ NO (2) ↓ TNF-α, IL-1β	[88]
Plant	Echinacoside	In vitro and in vivo studies	*Cistanche tubulosa* (Schrenk) Hook. f. II.	(1) LC-540, TM3 (2) SD rats	(1) 5, 10 μM (2) 160, 320 mg/kg; 6 weeks	(1) Increase of cell viability (2) Increase of sperm number, sperm motility, seminiferous tubule thickness Decrease of sperm abnormality	(1) LC-540, TM3: ↓ Superoxide anion, LC-540: ↑ StAR, CYP11A1, CYP17A1, HSD17β3 ↓ RAGE, NF-*κB*, H_2_O_2_ (2) ↑ LH, KISS1, SIRT1, GPR54, SOCS -3, SOD, CAT ↓ NO, TNF-α, IL-6, superoxide, MDA	[22]

NO, nitric oxide; TNF-α, tumor necrosis factor α; IL-1β, interleukin 1 beta; CYP17A1, Cytochrome P450 17A1; CYP11A1, cytochrome P450 11A1; HSD17β3, hydroxysteroid dehydrogenase 17β3; RAGE, receptor for advanced glycation end products; NF-*κB*, nuclear factor kappa-light-chain-enhancer of activated B cells; H_2_O_2_, hydrogen peroxide; LH, luteinizing hormone; Kiss1, kisspeptin 1; SIRT1, sirtuin 1; GPR 54, G protein-coupled receptor; SOCS-3, suppressor of cytokine signaling 3; SOD, superoxide dismutase; CAT, catalase; IL-6, interleukin 6.

**Table 4 antioxidants-09-00957-t004:** Animal derived natural products and male infertility.

Classification	Compound/Extract	Source	Cell Line/Animal Model	Dose; Duration	Efficacy	Mechanism	Reference
Insect	Drone milk	*Apis mellifera*	SD rats	110 mg/kg; 5, 10 days	Increase of weight of androgen-sensitive organs (glans penis, seminal vesicle, muscles)	↑ Testosterone, SLAP	[89]
Animal	Gelam Honey	*Apis mellifera*	SD rats	1.0 mL/100 g; 60 days	Increase of fertility	↑ Fructose	[21]
Insect	Hydroethanolic extract of Indian propolis	*Apis* *mellifera*	Swiss albino mice	400 mg/kg; 4 weeks	Increase of testis weight, sperm count, total motility, spermatozoa with normal head morphology, spermatozoa with normal DNA, number of tubules with complete spermatogenesis, diameter of seminiferous tubule, number of germ cells Decrease of sperm DNA damage, chromatin immaturity, apoptosis in spermatogonial germ cell	↑ Testosterone, GSH, CAT ↓ MDA, RAD51	[90]
Animal	Spermaurin	Scorpion *Scorpio maurus palmatus*	(1) Bovine sperm (2) Monkey sperm (3) Mouse oocytes	(1) dilution 1/20; 10 min (2) dilution 1/40; 10 min (3) dilution 1/40; 4 h	Improvement of sperm motility		[91]

SD, Sprague Dawley; SLAP, spot14-like androgen-inducible protein; TBRAS, thiobarbituric acid-reactive substances; LDH, lactic dehydrogenase; GSH, glutathione; CAT, catalase; MDA, malondialdehyde.

**Table 5 antioxidants-09-00957-t005:** Fungus derived natural product and male infertility.

Classification	Compound/Extract	Source	Cell Line/Animal Model	Dose; Duration	Efficacy	Mechanism	Reference
Fungi	*Antrodia cinnamomea* ethanol extract	*Antrodia cinnamomea* Chang.	SD rats	385, 770, 1540 mg/kg; 5 weeks	Increase of total sperm count, motility rate Decrease of abnormal sperm count, DNA damage in sperm	↑ LH, testosterone, StAR, CYP11A1, 17β-HSD, SOD ↓ RAGE, GRP-78, H_2_O_2_, NO, MDA	[92]

SD, Sprague Dawley; LH, luteinizing hormone; StAR, steroidogenic acute regulatory; CYP11A1, cytochrome P450 11A1; 17β-HSD, 17β-hydroxysteroid dehydrogenase; SOD, superoxide dismutase; RAGE, receptor for advanced glycation end products; GRP-78, glucose-regulated protein-78; H_2_O_2_, hydrogen peroxide; NO, nitric oxide; MDA, malondialdehyde.

**Table 6 antioxidants-09-00957-t006:** Formula and male infertility.

Name of Formula	Source	Cell Line/Animal Model	Dose; Duration	Efficacy	Mechanism	Reference
Gosha-jinki-gan decoction	*Rehmanniae radix*, *Achyranthis radix*, *Corni officinalis* Sieb., *Dioscoreae opposita* Thunb., *Plantaginis semen*, *Alismatis orientale* Juzepzuk., *Poria Cocos* Wolf., *Moutan Radicis* Cortex., *Cinnamomum verum* J.S. Presl., *Aconitum napellus*	C57BL/6J mice	5.4%; 150 days	Increase of testis weight, epididymis spermatozoa count, fertility rate, normal seminiferous tubule appearance	↓ Caspase-8, TLR2, TLR4, F4/80, Fas	[93]
MOTILIPERM	*Morinda officinalis* How, *Allium cepa* L., *Cuscuta chinensis* Lamark	SD mice	100, 200 mg/kg; 4 weeks	Increase of left testis weight, sperm motility, sperm count, epididymis motility, vas deferens count, epididymis count, spermatogenic cell density Decrease of damage of seminiferous tubule	↓ MDA, testosterone GRP-78, p-IRE1α, p-JNK	[94]
Saikokaryukotsuboreito	*Bupleurum falcatum* L., *Pinellia ternata* Breitenbach, *Cinnamon verum* J.Presl, *Poria cocos* Wolf, *Scutellaria baicalensis* Georgi, *Zizyphus jujuba* var. inermis, *Crassostrea gigas*., *Panax ginseng* C. A. Mey., *Rinoceros* spp., *Rheum rhabarbarum*. L., *Zingiber officinale* Roscoe	C57BL/6 mice	300 mg/kg; 30 days	Increase of sperm number, sperm motility	↑ Testosterone, intertesticular testosterone, SYCP3	[95]
Shengjing capsule	*Cornu cervi panto trichum*, *Cordyceps sinensis*, *Polygonatum kingianum* Coll. Et Hermsl, *Panax ginseng* C. A. mey, *Cuscuta chinesnsis* Lam., *Lycium barbarum* L., *Astragalus complanatus* R. Brown, *Epimedium wushanense* TS Ying, *Cortex eucommiae*, *Rosae laevigata* Michx., *Verbena officinalis* L., *Rubus chingii* Hu, *Curculigo orchioides* Gaertner, *Psoralea corylifolia* L.	SPF Wistar mice	0.45, 0.90, 1.80 g/kg; 56 days	Increase of sperm motility, normal sperm morphology, normal DNA fragmentation rate Decrease of abnormal sperm morphology, abnormal DNA fragmentation rate	↑ GSH-PX, MDA, SOD, testosterone	[8]
Wuzi Yanzong pill	*Lycium barbarum* L., *Cuscuta chinensis* Lam., *Rubus chingii* Hu. *Schizandra chinensis* (Turcz.) Baill., *Plantago asiatica* L.	Kunming mice	1.0 g/kg; 21 days	Increase of testis weight, sperm count, sperm motility	↑ Testosterone, total antioxidant status, PCNA, ↓ MDA, total oxidant status, OSI	[30]
Yi Shen Jian Pi	*Cuscuta chinensis*, Lycium spp., Schisandra spp., Codonopsis pilosula, Astragalus membranaceus, Citrus reticulate, Bupleurum spp., Cimicifuga foetida, Ligusticum chuanxiong, Carthamus tinctorius	BALB/c mice	1.35, 2.70 mg/kg; 4 weeks	Increase of sperm quality		[96]

TLR2, toll like receptor 2; TLR4, toll like receptor; F4/80, EGF-like module-containing mucin-like hormone receptor-like 1; GRP-78, glucose-regulated protein -78; p-IRE1-reinositol requiring kinase 1α; p-JNK, phosphorylated c-Jun N-terminal kinase; SYCP3, synaptonemal complex protein 3; GSH-Px, glutathione peroxidase; SOD, superoxide dismutase; MDA, malondialdehyde; PCNA, proliferating cell nuclear antigen; OSI, oxidative stress index.

**Table 7 antioxidants-09-00957-t007:** Plant derived natural products and female infertility (in vitro studies).

Classification	Compound/Extract	System	Source	Cell Line/Animal Model	Dose; Duration	Efficacy	Mechanism	Reference
Plant	*Evodiae Fructus* extract	In vitro	*Evodia rutaecarpa* Benth	CHO-K1, COV434	10, 50, 100, 300, 500 µg/mL; 24 h	Decrease of the ovotoxicity	↑ mTOR, GSK-3β	[28]
Plant	*Perilla frutescens*	In vitro	*Perilla frutescens* var. acuta Kudo	Ishikawa, JAr	50 μg/mL; 48 h	Decrease of implantation failure	↑ LIF, Integrin β3, β5	[15]

CHO-K1, Chinese hamster ovary cells; COV434, human ovarian granulose cells; LIF, leukemia inhibitory factor.

**Table 8 antioxidants-09-00957-t008:** Plant derived natural products and female infertility (in vivo studies)

Classification	Compound/Extract	System	Source	Cell Line/Animal Model	Dose; Duration	Efficacy	Mechanism	Reference
Plant	*Anthocleista schweinfurthii* aqueous extract	In vivo	*Anthocleista schweinfurthii*	Wistar rats	200, 300, 400 mg/kg; 28 days	Decrease of oxidative stress in brain	↑ Glutathione ↓ Malondialdehyde	[97]
Plant	*Astragalus mongholicus* extract	In vivo	*Astragalus mongholicus*	SPF/ICR mice	5%; 56 days	Decrease of estrogen-dependent endometrial hyperplasia and ovarian dysfunction	↑ PPARs, mDECR, estradiol	[35]
Plant	Barley Date palm fruits	In vivo	*Hordeum vulgare* *Phoenix dactylifera*	*Rattus rattus*	10%; 120 days	Increase of the ovarian function and protection from high cholesterol diet	↑ CAT, SOD, GST ↓ MDA	[98]
Plant	Cinnamon powder	In vivo	*Cinnamomum verum*	C57BL/6 mice	10 mg/100 g; 20 days	Increase of insulin resistance	↑ FSH, IGFBP-1 ↓ Insulin, IGF-1	[99]
Plant	*Cyperus rotundus* Water extract	In vivo	*Cyperus rotundus*	C57BL/6 mice	31.68 mg/kg; 7 days	Increase of cell adhesion and implantation of blastocysts	↑ LIF, LIF-dependent integrin αV, β3, β5	[12]
Plant	*Eucalyptus robusta* leaf extraxct	In vivo	*Eucalyptus robusta*	Wistar rats	25 mg/kg; 5 days	Decrease of endometritis	↑ COX-1, COX-2 ↓ TLR-4, TLR-9, Myleoperoxidase, iNOS, NO, SAA	[31]
Plant	*Ficus asperifolia* aqueous extract	In vivo	*Ficus asperifolia* (L) Hook. Ex Miq (Moraceae)	Wistar rats	100 mg/kg once a day; 1 week (set I) or 4 weeks (set II)	Increase of conducive condition maintenance	↑ HDL cholesterol ↓ Total plasma cholesterol, LDL cholesterol	[100]
Plant	*Ficus vogelii* aqueous extract	In vivo	*Ficus vogelii*	Wistar rats	100, 300 mg/kg; 21 days	Decrease of lead reproductive toxicity	↓ SOD	[13]
Plant	*Milicia excelsa* Aqueous extract	In vivo	*Milicia excelsa*	Wistar rats	14, 7, 140 mg/kg; 7, 15 days	Decrease of the problems of amenorrhea	↑ FSH, Estradiol	[18]
Plant	*Schisandra chinensis* extract	In vivo	*Schisandra chinensis*	SD rats	200 mg/kg; 7 days		↓ PRL	[101]
Plant	*Senecio biafrae* (Oliv. and Hiern) J. Moore aqueous extract	In vivo	*Senecio biafrae* (Oliv. and Hiern) J. Moore	Albino Wistar rats	8, 32, 64, 128 mg/kg; 20 days	Increase of the fertility parameters, uterine weight	↑ FSH, LH, estradiol, progesterone	[20]
Plant	Soy isoflavones	In vivo	*Glycine max*	*Rattus norvegicus albinus*	150 mg/kg; 30 days	Increase of insulin sensitivity	↑ SLRP	[101]

VCD, 4-vinylcyclohexene diepoxide; LIF, leukemia inhibitory factor; PCOS, polycystic ovary syndrome; LH, luteinizing hormone; FSH, follicle-stimulating hormone; LDL, low-density lipoprotein, HDL, high-density lipoprotein; MPO, myeloperoxidase; iNOS, inducible nitric oxide synthase; SAA, serum amyloid A; SOD, superoxide dismutase; DHEA, dehydroepiandrosterone; PCOS, polycystic ovary syndrome; FSH, follicle-stimulating hormone; IGF-1, insulin-like growth factor 1; IGFBP-1, insulin-like growth factor binding protein-1; CAT, catalase; GST, glutathione S-transferase; MDA, malondialdehyde; SLRP, small leucine-rich proteoglycans; PPAR, peroxisome proliferator-activated receptors; mDECR, mitochondrial 2,4-dienoyl-CoA reductase.

**Table 9 antioxidants-09-00957-t009:** Animal derived natural products and female infertility.

Classification	Compound/Extract	Source	Cell Line/Animal Model	Dose; Duration	Efficacy	Mechanism	Reference
Animal	Royal jelly	*Apis mellifera*	Wistar rats	100, 200, 400 mg/kg; 14 days	Increase of ovarian hormones and folliculogenesis	↑ FRAP, progesterone, estradiol ↓ NO	[102]

FRAP, ferric reducing antioxidant power assay; NO, nitric oxide.

**Table 10 antioxidants-09-00957-t010:** Clinical studies.

Gender	Classification	Compound/Extract	Source	Cell Line/Animal Model	Dose; Duration	Efficacy	Mechanism	Reference
Male	Plant	Bajijiasu	*Morinda officinalis* F.C.	Human sperm DNA	0.50, 1.00, 2.00 mg/mL; 45 min		↑ CAT, GSH-Px, SOD ↓ MDA	[6]
Male	Plant	Inositol	Wholemeal cereals, citrus fruits, brewer’s yeast	Seminal fluid from healthy subjects and from subjects with oligoasthenoteratospermia	2 mg/mL; 30 min, 1, 2 h	Increase of spermatozoa intermediate tract thickness Decrease of amorphous materials, damage in mitochondrial cristae		[103]
Male	Animal	Spermaurin	*Sorpio maurus plamatus*	Human sperm	Dilution 1/40; 10 min	Increase of sperm motility		[91]
Male	Plant	*Tribulus terrestris* extract	*Tribulus terrestris* Linn.	Human sperm	40, 50 mg/mL; 120 min	Increase of total sperm motility, number of progressive motile spermatozoa, curvilinear velocity		[104]
Male	Plant	*Tribulus terrestris* extract	*Tribulus terrestris* Linn.	Human sperm	20, 40, 50 μg/mL; 120 min	Increase of sperm viability, progressive motile spermatozoa		[105]
Male	Insect	Chilean propolis ethanol extract	*Apis mellifera*	Normozoospermic semen samples	6, 12, 25 μg/mL; 1 h	Decrease of intracellular oxidants in spermatozoa and DNA damage	↓ TBARS, LDH	[2]
Female	Plant	Low-dose aspirin in combination with Tiao Jing Cu Yun pills	Aspirin	PCOS patients	Enteric-coated aspirin tablets: 100 mg/day; until the day of ovulation	Increase of perifollicular artery blood flow, oocyte quality, rate of clinical pregnancy	↑ FSH, estradiol, progesterone	[106]
Female	Plant	Cinnamon supplements	*Cinnamomum cassia*	45 women with PCOS	1.5 g/d; 6 months	Increase of menstrual cyclicity, insulin resistance	↑ Insulin sensitivity, HOMA ↓ QUICK-I	[107]
Female	Plant	Fenugreek seed extract	*Trigonella foenumgraecum* L.	50 premenopausal women diagnosed with PCOS	Two capsules of 500 mg; 90 days	Decrease of ovarian volume, the number of ovarian cysts	↑ LH, FSH	[108]

CAT, catalase; SOD, superoxide dismutase; GSH-Px, glutathione peroxidase; MDA, malondialdehyde; TBRAS, thiobarbituric acid reactive substance; LDH, lactate dehydrogenase; FSH, follicle stimulating hormone; HOMA, homeostasis model assessment; QUICK-I, quantitative *insulin sensitivity* check index; LH, luteinizing hormone.

**Table 11 antioxidants-09-00957-t011:** Natural products with adverse effects.

Classification	Compound/Extract	Source	Cell Line/Animal Model	Dose; Duration	Efficacy	Mechanism	Reference
Plant	Olea hydro-alcoholic extract	*Olea europaea*	SD rats	50, 150, 450 mg/kg; 48 days	Decrease of weight of testicle, seminal vesicle, testosterone hormone, sperm count, motility	↓ LH, 17 β-hydroxy steroid hydrogenase	[109]
Plant	Permethrin		Wistar rats	35 mg/kg; 60 days	Decrease of Leydig cells		[110]
Plant	Red cultivar *Allium cepa* methanol extract	*Allium cepa* Linn.	Wistar rats	100, 200, 40, 800, 1200 mg/kg; 14 days	Decrease of sperm concentration, sperm motility, sperm mass activity		[111]
Plant	*Ricinus communis* L. aqueous extract	*Ricinus communis* L.	Human sperm	100, 200, 300 mg/mL; 30 min	Decrease of sperm motility	↓ NCD, HOS	[112]
Plant	*Spondias Mombin* L. ethanol extract	*Spondias Mombin* L.	Wistar strain rats	250, 500 mg/kg; 8 weeks	Decrease of spermatogenesis process	↓ Androgen	[113]
Plant	*Tripterygium glycoside* extract	*Tripterygium wilfordii* Hook. F. (GTW)	Female SD rats	37.8, 94.5 mg/kg; 90 days	Decrease of metestrus phase	↓ ER-α in hypothalamus	[114]
			Male SD rats	37.8, 94.5 mg/kg; 90 days	Increase of immature sperm, sperm abnormality rate	↓ AR in testis and epididymis	

LH, luteinizing hormone; NCD, nuclear chromatin decondensation test; HOS, hypo-osmotic swelling test; ER-α, estrogen receptor alpha; AR, androgen receptor.

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
