# Peer review of "Role of Antioxidant Natural Products in Management of Infertility: A Review of Their Medicinal Potential"

_antioxidants, 2020, doi:10.3390/antiox9100957_

Round 1

Reviewer 1 Report

Comments to the Authors

Oxidative stress is caused by an imbalance between ROS and antioxidants, which plays a significant role in the pathophysiology of many human diseases. However, there is extensive evidence highlighting the role of oxidative stress in male infertility due to raised levels of sperm DNA fragmentation and abnormal semen parameters. Oxidative stress may also lead to embryo fragmentation and formation of numerous developmental abnormalities, and is regarded to be one of the important reasons of spontaneous and recurrent miscarriage. The use of antioxidants is a potential therapeutic option to reduce ROS and improve semen quality and reduced male and female fertility.

The subject of the manuscript is interesting. The authors move very important topics on several beneficial natural products possessed antioxidant properties and androgenic activities on productive factors and hormones, since the lot of factors determine the deleterious effects of environmental toxicants on the functioning of the female and male reproductive system leading to their damages and affect following generations.

The manuscript is recommended for publication in the present form.

Author Response

We appreciate editors and reviewers for critical comments to improve the quality of our manuscript (antioxidants-920616), titled “Role of antioxidant natural products in management of infertility: A review of their medicinal potential”. We earnestly responded to the raised comments point by points.

Oxidative stress is caused by an imbalance between ROS and antioxidants, which plays a significant role in the pathophysiology of many human diseases. However, there is extensive evidence highlighting the role of oxidative stress in male infertility due to raised levels of sperm DNA fragmentation and abnormal semen parameters. Oxidative stress may also lead to embryo fragmentation and formation of numerous developmental abnormalities, and is regarded to be one of the important reasons of spontaneous and recurrent miscarriage. The use of antioxidants is a potential therapeutic option to reduce ROS and improve semen quality and reduced male and female fertility.

The subject of the manuscript is interesting. The authors move very important topics on several beneficial natural products possessed antioxidant properties and androgenic activities on productive factors and hormones, since the lot of factors determine the deleterious effects of environmental toxicants on the functioning of the female and male reproductive system leading to their damages and affect following generations.

The manuscript is recommended for publication in the present form.

(Response): Thank you very much for your comments. We have the infertility problems in Korea like other countries. We hope this study helps the researchers and patients who are dealing with this sad disease.

Again we appreciate reviewers and editors for their kind and careful comments for improving the quality of our manuscript and also sincerely hope we address our responses well to the raised comments and our revised manuscript would be accepted for publication in your journal soon.

With kind regards,

Prof. Bonglee Kim, M.D, Ph.D.

Department of Pathology, College of Korean Medicine, Kyung Hee University

1 Hoegi-dong, Dongdaemun-ku, Seoul 130 -701, South Korea

E-mail: bongleekim@khu.ac.kr

Tel; +82-2-961-9217, Fax; +82-2-961-9217

Reviewer 2 Report

The review details about the role of natural substances, extracts, and natural formulations in the field of infertility. The review provided a thorough description of the studies in this area but there are some issues to be addressed.

  1. Abstract: Explain in detail the natural products showing antioxidant properties and, at the same time, the natural products showing adverse effects both in male and female.
  2. In the paragraph one, there is a lack of information concerning the role of ROS and oxidative stress (OS) in the female reproductive system and about the hypothalamic-pituitary-gonadal axis alterations in the context of male infertility. Please, add these notions.
  3. Paragraph 1.3: Concerning the role of plant extracts in the context of reproductive systems, the authors focused only on the male side. Please, introduce this concept also in the context of female reproductive system.
  4. Although human studies cited in the text are few, it would be better to distinguish in different paragraphs and tables the experimental studies using animal models and the human studies in order to facilitate the understanding of the experiments.
  5. Paragraph 2.3. Are there natural products with adverse effects on the female reproductive system? Please, add this important information.

Author Response

We appreciate editors and reviewers for critical comments to improve the quality of our manuscript (antioxidants-920616), titled “Role of antioxidant natural products in management of infertility: A review of their medicinal potential”. We earnestly responded to the raised comments point by points.

The review details about the role of natural substances, extracts, and natural formulations in the field of infertility. The review provided a thorough description of the studies in this area but there are some issues to be addressed.

Abstract: Explain in detail the natural products showing antioxidant properties and, at the same time, the natural products showing adverse effects both in male and female.

(Response): Thank you for your comments. Detail of natural products showing antioxidant efficacies and adverse effects in both sexes are added. However there is limitation of words number in abstract (less 200 words), the part is briefly added.

In the paragraph one, there is a lack of information concerning the role of ROS and oxidative stress (OS) in the female reproductive system and about the hypothalamic-pituitary-gonadal axis alterations in the context of male infertility. Please, add these notions.

(Response): Thanks. The role of ROS and OS in the female reproductive system and the hypothalamic-pituitary-gonadal axis alterations are added in the paragraph one.

Paragraph 1.3: Concerning the role of plant extracts in the context of reproductive systems, the authors focused only on the male side. Please, introduce this concept also in the context of female reproductive system.

(Response): Thank you for the kind comments. We added the concept on the female side in 1. Infertility section, because paragraph 1.3 is merged into paragraph 1.

Although human studies cited in the text are few, it would be better to distinguish in different paragraphs and tables the experimental studies using animal models and the human studies in order to facilitate the understanding of the experiments.

(Response): Thank you for the comments. All human studies are distinguished in “2.3. Natural products and clinical studies” and table 9 for better understanding.

Paragraph 2.3. Are there natural products with adverse effects on the female reproductive system? Please, add this important information

(Response): Thanks. We found one more study about natural product with adverse effects on the female metestrus phase. There was just one study which is conforming to our criteria. The information of the study is added both in 2.4. Natural products with adverse effects and table 10.

Again we appreciate reviewers and editors for their kind and careful comments for improving the quality of our manuscript and also sincerely hope we address our responses well to the raised comments and our revised manuscript would be accepted for publication in your journal soon.

With kind regards,

Prof. Bonglee Kim, M.D, Ph.D.

Department of Pathology, College of Korean Medicine, Kyung Hee University

1 Hoegi-dong, Dongdaemun-ku, Seoul 130 -701, South Korea

E-mail: bongleekim@khu.ac.kr

Tel; +82-2-961-9217, Fax; +82-2-961-9217

Reviewer 3 Report

This is a review about the mechanisms, use, and potential applications of natural products on male and female infertility.

The review starts from a list of factors affecting infertility, as oxidative stress, inflammation, and pituitary hormones. Then most of the ms is dedicated to the review of literature on natural products from plants, animals and fungi affecting male and female infertility.

The study could be of interest to numerous readers but, in the reviewer's opinion, is not suitable for publication as presented.

The ms needs an extensive English revision, and a careful check of typing and formatting mistakes (just for exampe: pregnelone instead of Pregnenolone, Fig.1; leydig instead of Leydig).

Subchapter 1.3 and 2 share the same title. A different title and subject should be proposed.

lines 78-80 (and subsequent): "Estrogen promote proliferation, suppress apoptosis of granulosa cells Oocyte-derived paracrine factors (ODPFs)". This sentence, and also the one following this, is unclear and should be rewritten.

The subchapter 2.1.1 is difficult to read and excessively long. It is suggested to separate it in different subheadings, one for each natural compound.

Line 139: what do you mean that Adryamicin "recovered seminiferous tubules"? please, explain the parameter used to evaluate recovery.

Lines 200-201: "The administration of the extract to cadmium induced Wistar rats resulted in an increase of mean sperm number and a regulation of testosterone and LPO levels". Do you mean "cadmium exposed"?

Lines 934-936: "Mitomycin C, cyclophosphamide, cisplatin and Adriamycin have been demonstrated as applicable reproductive agents to reduce the negative effects of chemotherapy". Are not them anticancer agents? please explain how they can reduce the negative effect of chemotherapy

Author Response

We appreciate editors and reviewers for critical comments to improve the quality of our manuscript (antioxidants-920616), titled “Role of antioxidant natural products in management of infertility: A review of their medicinal potential”. We earnestly responded to the raised comments point by points.

This is a review about the mechanisms, use, and potential applications of natural products on male and female infertility. The review starts from a list of factors affecting infertility, as oxidative stress, inflammation, and pituitary hormones. Then most of the ms is dedicated to the review of literature on natural products from plants, animals and fungi affecting male and female infertility. The study could be of interest to numerous readers but, in the reviewer's opinion, is not suitable for publication as presented.

The ms needs an extensive English revision, and a careful check of typing and formatting mistakes (just for exampe: pregnelone instead of Pregnenolone, Fig.1; leydig instead of Leydig).

(Response): Thank you for the comments. Whole MS is carefully checked again including pregnelone and leydig.

Subchapter 1.3 and 2 share the same title. A different title and subject should be proposed.

(Response): Sorry for the confusion caused. The 1.3 and 2 are merged to make them natural.

lines 78-80 (and subsequent): "Estrogen promote proliferation, suppress apoptosis of granulosa cells Oocyte-derived paracrine factors (ODPFs)". This sentence, and also the one following this, is unclear and should be rewritten.

(Response): Sorry for the confusion caused. The sentences are revised as “Estrogen affects granulosa cells by promotion of proliferation, suppression of apoptosis, and augmentation of FSH effects.”

The subchapter 2.1.1 is difficult to read and excessively long. It is suggested to separate it in different subheadings, one for each natural compound.

(Response): You are right. 2.1.1 was too long. Each subheading for each natural compound could be too many subheadings, we divided it into three subheading including “2.1.1.1. in vitro studies”, “2.1.1.2. in vivo studies”, “2.1.1.3. in vitro and in vivo studies”. Also table 1 is separated to table 1, 2, and 3.

Line 139: what do you mean that Adryamicin "recovered seminiferous tubules"? please, explain the parameter used to evaluate recovery.

(Response): Sorry for the confusion. The sentence is revised as “The aged garlic extract recovered the adriamycin induced testicular changes including low testis weight, low sperm count, low motility, thick irregular basal lamina of seminiferous tubules, and sperm abnormality.”

Lines 200-201: "The administration of the extract to cadmium induced Wistar rats resulted in an increase of mean sperm number and a regulation of testosterone and LPO levels". Do you mean "cadmium exposed"?

(Response): Yes, revised.

Lines 934-936: "Mitomycin C, cyclophosphamide, cisplatin and Adriamycin have been demonstrated as applicable reproductive agents to reduce the negative effects of chemotherapy". Are not them anticancer agents? please explain how they can reduce the negative effect of chemotherapy

(Response): Sorry for the confusion caused. The sentence is revised as “Several natural products in this review have been demonstrated as applicable reproductive agents to reduce the negative effects of chemotherapy such as mitomycin C, cyclophosphamide, cisplatin and adriamycin”

Again we appreciate reviewers and editors for their kind and careful comments for improving the quality of our manuscript and also sincerely hope we address our responses well to the raised comments and our revised manuscript would be accepted for publication in your journal soon.

With kind regards,

Prof. Bonglee Kim, M.D, Ph.D.

Department of Pathology, College of Korean Medicine, Kyung Hee University

1 Hoegi-dong, Dongdaemun-ku, Seoul 130 -701, South Korea

E-mail: bongleekim@khu.ac.kr

Tel; +82-2-961-9217, Fax; +82-2-961-9217

Round 2

Reviewer 2 Report

The authors properly addressed this reviewer comments

Reviewer 3 Report

The ms was extensively revised, according to the referee's suggestion and can be now suitable for publication